# Identifying biological perturbation targets through causal differential networks

Menghua Wu [1]   Umesh Padia [1]   Sean H. Murphy [1]   Regina Barzilay [1]   Tommi Jaakkola [1]

## Abstract

Identifying variables responsible for changes to a biological system enables applications in drug target discovery and cell engineering. Given a pair of observational and interventional datasets, the goal is to isolate the subset of observed variables that were the targets of the intervention. Directly applying causal discovery algorithms is challenging: the data may contain thousands of variables with as few as tens of samples per intervention, and biological systems do not adhere to classical causality assumptions. We propose a causality-inspired approach to address this practical setting. First, we infer noisy causal graphs from the observational and interventional data. Then, we learn to map the differences between these graphs, along with additional statistical features, to sets of variables that were intervened upon. Both modules are jointly trained in a supervised framework, on simulated and real data that reflect the nature of biological interventions. This approach consistently outperforms baselines for perturbation modeling on seven single-cell transcriptomics datasets. We also demonstrate significant improvements over current causal discovery methods for predicting soft and hard intervention targets across a variety of synthetic data.[1]

## 1. Introduction

Cells form the basis of biological systems, and they take on a multitude of dynamical states throughout their lifetime. In addition to natural factors like cell cycle, external perturbations (e.g. drugs, gene knockdown) can alter a cell's state. While perturbations can affect numerous downstream variables, identifying the root causes, or *targets*, that induce these transitions has vast therapeutic implications, from cellular reprogramming (Cherry & Daley, 2012) to mechanism of action elucidation (Schenone et al., 2013).

Machine learning methods have primarily been designed to infer the effects of perturbations on cells, with the goal of generalizing to unseen perturbations (Roohani et al., 2023), or unseen cell distributions (Lotfollahi et al., 2023; Bunne et al., 2023). In principle, these models can be used within an active learning framework to discover intervention targets (Huang et al., 2024; Zhang et al., 2023a). However, the number of candidate sets scales exponentially with the number of variables under consideration, and there are limited training data for combinatorial perturbations ($< 150$ pairs in Norman et al. (2019)). Thus, this option may be impractical for larger search spaces.

Alternatively, some methods predict targets by comparing pairs of "before" and "after" cell states, similar to this paper. They do so by attributing observed differences to sparse, mechanistic changes. Specifically, relationships between variables are represented through a graph, e.g. a gene regulatory network (Babu et al., 2004). Existing methods often use data-mined relations as priors for this graph, and search for modifications to the set of edges that best explain the interventional data (Cosgrove et al., 2008; Gonzalez et al., 2024). However, biological knowledge graphs are incomplete and noisy priors. They contain heterogeneous, potentially conflicting information, since the reported findings were observed under diverse contexts (Ashburner et al., 2000). Moreover, different cell populations may exhibit distinct gene regulatory behavior (Huttlin et al., 2021), leading to incomplete knowledge, especially of less well-characterized systems.

In this paper, we propose causal differential networks (CDN): a causality-inspired approach to identify variables that drive desired shifts in cell state, while estimating their mechanistic structure directly from data. Given a pair of observational and interventional datasets, we train a causal structure learner to predict causal graphs that could have generated each dataset. The pair of graphs is input to an attention-based classifier, which predicts whether each variable was subject to intervention. To address the challenges of data

---

[1]Department of Computer Science, Massachusetts Institute of Technology, Cambridge, MA, USA. Correspondence to: Menghua Wu <rmwu@mit.edu>.

*Proceedings of the $42^{nd}$ International Conference on Machine Learning*, Vancouver, Canada. PMLR 267, 2025. Copyright 2025 by the author(s).

[1]Code is available at https://github.com/rmwu/cdn

noise and sparsity, and to simulate the structure of biological interventions, these models are trained jointly in a supervised (amortized) framework (Ke et al., 2023; Petersen et al., 2023), over thousands of synthetic or real datasets.

We evaluate CDN on real transcriptomic data and synthetic settings. CDN outperforms the state-of-the-art in perturbation modeling (deep learning and statistical approaches), evaluated on the five largest Perturb-seq datasets at the time of publication (Replogle et al., 2022; Nadig et al., 2024) without using any external knowledge. Furthermore, CDN generalizes with minimal performance drop to unseen cell lines, which have different supports (genes), causal mechanisms (gene regulatory networks), and data distributions. On synthetic settings, CDN outperforms causal discovery approaches for estimating unknown intervention targets.

## 2. Background and related work

### 2.1. Problem statement

We consider a system of $N$ random variables $X_i \in X$, whose relationships are represented by directed graph $G = (V, E)$, where nodes $i \in V$ map to variables $X_i$, and edges $(i, j) \in E$ to relationships from $X_i$ to $X_j$. Samples from the data distribution $P_X$ are known as *observational*. We can perform *interventions* by assigning new conditionals

$$P(X_i \mid X_{\pi_i}) \leftarrow \tilde{P}(X_i \mid X_{\pi_i}), \tag{1}$$

where $\pi_i$ denotes the parents of node $i$ in $G$. *Hard* interventions remove all dependence between $X_i$ and $\pi_i$, while *soft* interventions maintain the relationship with a different conditional. We denote the joint *interventional* distribution as $\tilde{P}_X$. Given an observational dataset $D_{\text{obs}} \sim P_X$ and an interventional dataset $D_{\text{int}} \sim \tilde{P}_X$, our goal is to predict the set of nodes $I$ for which

$$P(X_i \mid X_{\pi_i}) \neq \tilde{P}(X_i \mid X_{\pi_i}), \forall i \in I. \tag{2}$$

In the context of molecular biology, a *perturbation* is a hard or soft intervention, and $I$ is the set of perturbation *targets*, where $\|I\| \geq 1$.

### 2.2. Modeling perturbations

Perturbation experiments result in datasets of the form $D_{\text{obs}}, \{(I^k, D_{\text{int}}^k)\}$, where $D_{\text{obs}} \sim P_X$, $D_{\text{int}}^k \sim \tilde{P}_X^k$. In this paper, we focus on the *transcriptomic* data modality, where variables $X_i$ represent the levels of gene $i$; the number of variables $N$ ranges from hundreds to thousands; $D_{\text{obs}}$ contains thousands of samples; and each $D_{\text{int}}$ contains tens to hundreds of samples (Replogle et al., 2022).

Each experiment is conducted in a single *cell line*, which represents a distinct set of variables $X$, distribution $P_X$, and graph $G$. These three objects may share similarities across

cell lines, as certain genes or pathways are essential for survival. However, the empirical discrepancy can be quite large between experiments. For example, Nadig et al. (2024) reports that the median correlation in log-fold change of genes under the same perturbation, across two experiments in the same cell line, by the same lab, was 0.16.

Early efforts to map the space of cells measured the distribution of gene levels as a surrogate for cell state (Lamb et al., 2006). Diseases, drugs, or genetic perturbations that induce similar changes in this distribution were thought to act through similar mechanisms. However, it is intractable to map all combinations of starting cell states and perturbations through experiment alone. This bottleneck has led to two directions in the *in-silico* modeling of perturbations.

To reduce experimental costs, which scale with the number of cell lines and perturbations, one option is to infer the post-intervention distribution of cells. That is, given $D_{\text{obs}} \sim P_X$ and $I$, the goal is to infer $\tilde{P}_X$. Works such as scGen (Lotfollahi et al., 2019), CPA (Lotfollahi et al., 2023), Chem-CPA (Hetzel et al., 2022), and CellOT (Bunne et al., 2023) aim to generalize to new settings $P_X$, e.g. unseen cell lines or patients. Alternatively, models like GEARS (Roohani et al., 2023), graphVCI (Wu et al., 2023), AttentionPert (Bai et al., 2024), and GIM (Schneider et al., 2024) infer the effects of unseen perturbations. This setting is also a common application for single cell foundation models (Theodoris et al., 2023; Cui et al., 2024; Hao et al., 2024). These latter methods generalize by modeling the relationships between seen and unseen perturbations. For example, GEARS leverages literature-derived relationships in the Gene Ontology knowledge graph (Ashburner et al., 2000), and graphVCI learns to refine the gene regulatory relationships implied by ATAC-seq data (Grandi et al., 2022).

Approaches for inferring the effects of interventions do *not* explicitly predict targets, but if the space of perturbations $\mathcal{I}$ can be efficiently enumerated, these models are useful for comparing different $I$ (Zhang et al., 2023a; Huang et al., 2024). For example, given an oracle $\phi : D_{\text{obs}}, I \rightarrow \tilde{P}$, the $I^*$ associated with $\tilde{P}^*$ is

$$I^* = \arg \min_{I \in \mathcal{I}} d(\tilde{P}^*, \phi(D_{\text{obs}}, I)) \tag{3}$$

for some dissimilarity $d$. However, recent works have reported that the predictive ability of current models does not exceed that of naive baselines (Kernfeld et al., 2023; Ahlmann-Eltze et al., 2024; Märtens et al., 2024), reflecting that phenotype prediction is currently unsolved.

Perturbation datasets can also be used to train models for predicting targets of interest, as is the focus of this work. A common strategy is to relate observed changes (e.g. disease, gene levels) to mechanistic explanations (e.g. pathways), in which the targets are involved. To draw this connection, current approaches use external information such as

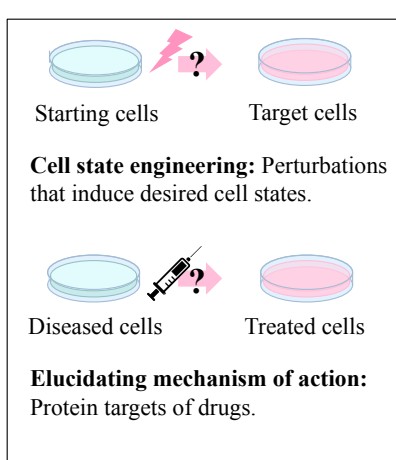 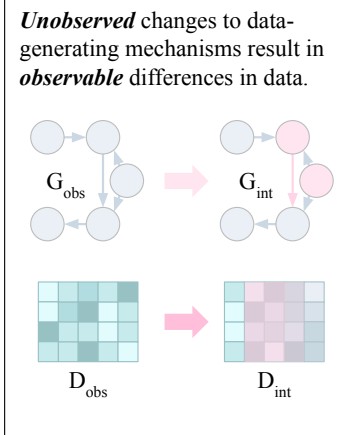 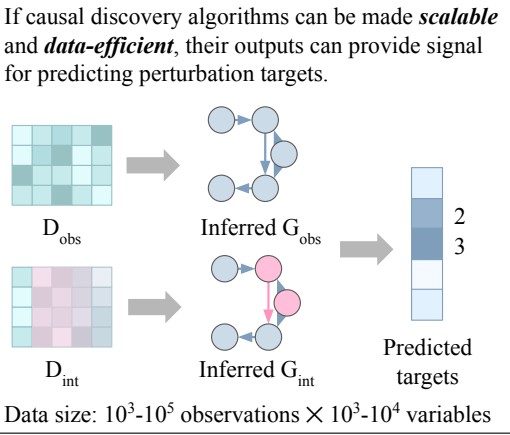

(A) Perturbation target prediction     (B) Causal interpretation     (C) Causality-inspired approach

Figure 1: (A) Biological applications. (B) Differences between observation dataset $D_{\text{obs}}$ and interventional dataset $D_{\text{int}}$ can be attributed to changes in the underlying causal mechanisms. (C) We use an amortized causal discovery model to predict $G_{\text{obs}}$ and $G_{\text{int}}$, whose hidden representations are input to a perturbation target classifier. Both modules are jointly trained.

knowledge graphs (Cosgrove et al., 2008; Gonzalez et al., 2024) and natural language (Roohani et al., 2025). However, these information are collected from highly inhomogenous sources (Ashburner et al., 2000), and it is difficult to quantify their correctness and completeness in each new setting. An alternative is to infer the mechanistic structure directly from data, either explicitly (di Bernardo et al., 2005) or as a model parameter (Noh & Gunawan, 2016). In this work, we follow the latter strategy, while allowing for a richer (non-linear) class of relationships.

### 2.3. Causal discovery

The task of inferring causal mechanisms from data is known as causal discovery, or causal structure learning (Spirtes et al., 2001). In the simplest setting, the goal is to infer $G$ from $D_{\text{obs}}$. The earliest works were discrete search algorithms, which operated over the combinatorial space of potential graphs (Spirtes et al., 1995; Tian & Pearl, 2001; Chickering, 2002; Shimizu et al., 2006). More recently, a number of works have reframed the task as a constrained optimization over continuous adjacency matrices (Zheng et al., 2018; Lachapelle et al., 2020; Sanchez et al., 2023).

If data from multiple environments are available, they can provide additional identifiability regarding the system (Jaber et al., 2020; Zhang et al., 2023b). Differences between these environments, e.g. intervention targets $I$, are either given as input or part of the inference task. As in the observational case, discovery algorithms can be categorized into discrete search (Hauser & Bühlmann, 2012; Ghassami et al., 2018; Ke et al., 2019; Huang et al., 2020; Mooij et al., 2020) or continuous optimization (Brouillard et al., 2020; Hägele et al., 2023). Additionally, Varici et al. (2022) and Yang et al. (2024) frame intervention target prediction as its own task, similar to this paper. However, the former strictly as-

sumes linearity, while the latter requires at least as many environments as $\|I\|$ (we only consider before and after perturbation). Finally, several works aim to infer differences between causal mechanisms (edges), rather than identifying the full causal structure (Wang et al., 2018; Belyaeva et al., 2021; Chen et al., 2023; Malik et al., 2024). Our work is also motivated by the idea that differences between summary statistics (e.g. regression coefficients in DCI) are informative of changes between systems. Thus, CDN can be viewed as implementing similar operations within an end-to-end, supervised framework.

### 2.4. Amortized causal inference

While the majority of causal discovery algorithms must be fit or run from scratch for each data distribution, amortized causal discovery algorithms have addressed this task as a supervised machine learning problem. Using large numbers of synthetic datasets, generated by synthetic graphs, a neural network is trained to map $D$ directly to $G$ (Li et al., 2020; Ke et al., 2023; Lorch et al., 2022; Petersen et al., 2023; Wu et al., 2025). Compared to traditional approaches, these algorithms trade explicit assumptions for implicit characteristics of the data simulation process (Montagna et al., 2024). These models are also fast and robust to noise. In this work, we simulate interventions that reflect biological perturbations, and our model architecture builds upon that of Wu et al. (2025), as it scales easily to hundreds of variables.

### 3. Methods

We introduce causal differential networks (CDN), a causality inspired method for predicting intervention targets. On a high level, CDN is motivated by the idea that differences in observations $D_{\text{obs}}$ and $D_{\text{int}}$ can be explained by the changes

to their underlying causal mechanisms (Figure 1B).

Our model is composed of two modules: a *causal structure learner* (Section 3.1), and a *differential network* classifier (Section 3.2). Given a single dataset $D$, the causal structure learner predicts a putative $G$. The pair of datasets $D_{\text{obs}}$ and $D_{\text{int}}$ yields graphs $G_{\text{obs}}$ and $G_{\text{int}}$, whose features are input to the differential network, to predict targets $I$ (Figure 1C). Both modules are trained jointly, supervised by classification losses for the graph and predicted targets. Finally, individual samples are known to be noisy in single-cell biology and are often analyzed in "pseudo-bulk" (Blake et al., 2003). Thus, instead of operating over raw data, we also represent datasets in terms of their summary statistics.

### 3.1. Causal structure learner

The causal structure learner takes as input dataset $D$ and outputs the predicted $G$. In principle, this module can be implemented with any data-efficient causal discovery model. Here, we based the architecture on SEA (Wu et al., 2025), as its runtime complexity does not explicitly depend on the number of samples (which can be large for $D_{\text{obs}}$).

**Input featurization** We represent dataset $D$ through two types of statistics. First, we compute global statistics $\rho \in \mathbb{R}^{N \times N}$ (pairwise correlation) over all variables. This statistic is an inexpensive way for the model to filter out edges that are not likely to occur. Second, we obtain estimates of local causal structure to orient edges, by running classical causal discovery algorithms over small subsets of variables. Specifically, we run the FCI algorithm $T$ times over subsets of $k$ variables (Spirtes et al., 1995), which are sampled proportional to their pairwise correlation. We denote the set of estimates as $E' \in \mathcal{E}^{k \times k \times T}$, where $\mathcal{E}$ is the space of possible edge types inferred by FCI (e.g. $\leftrightarrow, -, \rightarrow$). The size and number of local estimates, $k$ and $T$, are hyperparameters that trade off accuracy vs. runtime (Section 3.3).

To compare $\rho$ and $E'$ in the same space, we linearly project elements of $\rho$ from $\mathbb{R}$ to $\mathbb{R}^d$, and map categorical edge types into $\mathbb{R}^d$ using a learned embedding, i.e. like token embeddings in language models (Devlin et al., 2019). Thus, the dataset $D$ is summarized into a matrix of size $\mathbb{R}^{N \times N \times T \times d}$.

**Model architecture** Both the causal structure learner and differential network are implemented with a series of attention-based blocks. Since scaled dot-product attention scales quadratically as sequence length (Vaswani et al., 2017), we attend over one dimension at a time. For example, given a $N^2$ entries in an adjacency matrix, naive self-attention would cost $O(N^4)$ operations. Instead, we apply self-attention along all nodes in the "outgoing edge" direction, with the "incoming edges" as a batch dimension, followed by the opposite – resulting in two operations, each of cost $O(N \times N^2)$. Following Rao et al. (2021), we use

pre-layer normalization on each self-attention, followed by dropout and residual connections:

$$h \leftarrow h + \text{Dropout}(\text{Self-Attn}(\text{LayerNorm}(h))) \quad (4)$$

where $h \in \mathbb{R}^{\cdots \times d}$ denotes an intermediate hidden representation. This design takes after Transformer models for higher dimensional data in other domains, e.g. Axial Transformers for videos (Ho et al., 2019) and MSA Transformers for aligned protein sequences (Rao et al., 2021). A key difference is the graph modality: when making edge or node-level predictions, the model should be equivariant to node labeling. This is accomplished by adding randomly-permuted "positional" embeddings to each node and edge representation. For more details, please see Appendix A.

**Edge-level outputs** Let $h_{\text{obs}}, h_{\text{int}} \in \mathbb{R}^{N \times N \times d}$ denote the final layer representations of each dataset, after pooling over the $T$ dimension. These representations are input to the differential network. In addition, we introduce a graph prediction head,

$$z_{i,j} = \text{FFN}([h_{i,j}, h_{j,i}]) \qquad \text{(logits)} \quad (5)$$
$$P(E_{i,j} = e) = \sigma(z_{i,j})_e \qquad \text{(softmax)} \quad (6)$$

where $\sigma$ denotes softmax over the set of output edge types (forwards, backwards, or no edge). The graph prediction loss $\mathcal{L}_G$ is simply the cross entropy loss between the predicted and true edge types, summed over the entire graph.

### 3.2. Differential network

Given a pair of graph representations $h_{\text{obs}}, h_{\text{int}}$, the differential network predicts $I$, the set of intervention targets.

**Input featurization** In addition to pairwise representations $h$, we compute node-level statistics (mean, variance), which are mapped to $\mathbb{R}^d$ via a feed-forward network, and concatenated along "incoming edge" dimension. This results in graph representation $h' \in \mathbb{R}^{N \times (N+1) \times d}$.

We combine the graph representations either through concatenation or subtraction:

$$h_{\text{cat}} = [h'_{\text{obs}}, h'_{\text{int}}] \in \mathbb{R}^{N \times (N+1) \times 2d} \quad (7)$$
$$h_{\text{diff}} = h'_{\text{int}} - h'_{\text{obs}} \in \mathbb{R}^{N \times (N+1) \times d}. \quad (8)$$

The former can express richer relationships between the two graphs, while the latter is more efficient on large graphs.

**Model architecture** The differential network closely follows the architecture of the causal structure learner. However, since paired graph representation lacks the $T$ dimension (introduced by marginal estimates $E'$), the model only needs to attend over two "length" dimensions (rows and columns of adjacency matrix). After the attention layers, we mean over all "incoming edges" and linearly project to binary node-level predictions, where a normalized 1 indicates an intervention target, and 0 indicates otherwise.

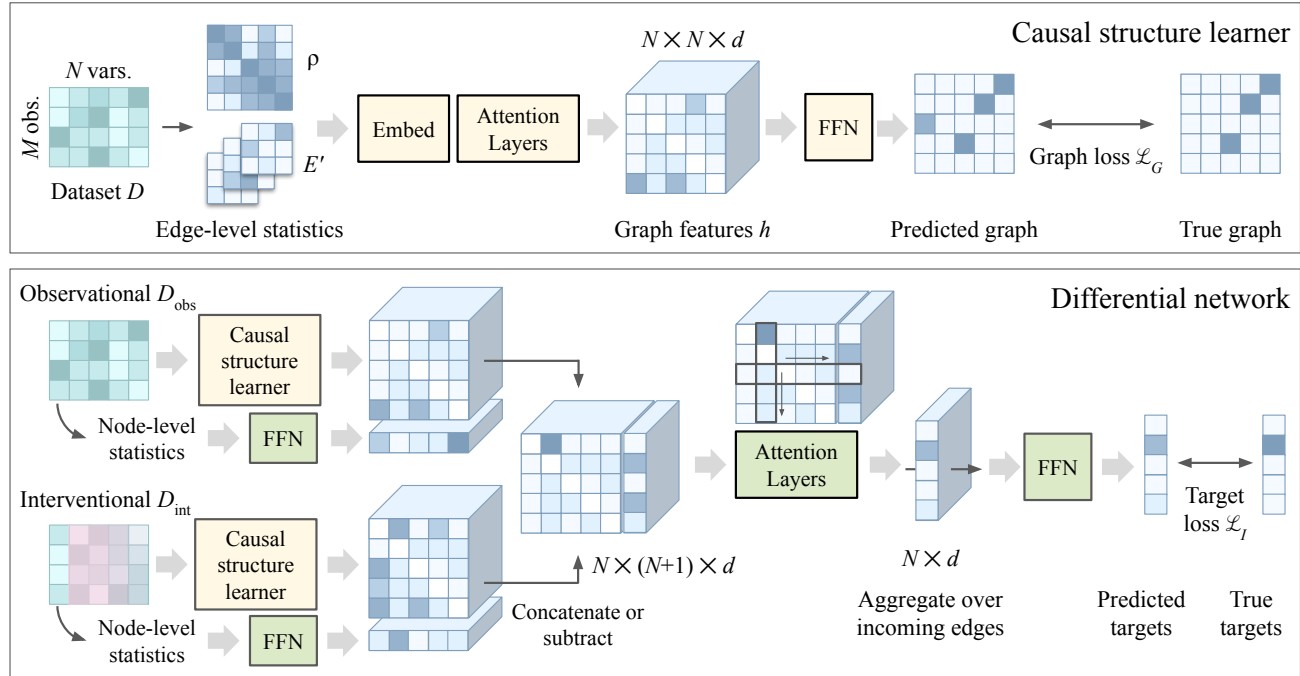

Figure 2: Our model is composed of a causal structure learner and a differential network. Learned components in yellow (causal structure learner) and green (differential network). Both modules are trained jointly to optimize $\mathcal{L} = \mathcal{L}_G + \mathcal{L}_I$.

The target prediction loss $\mathcal{L}_I$ is the binary cross entropy loss between the predicted and true targets, computed independently for each node. This is to accommodate interventions with multiple targets, e.g. drugs with off-target effects.

**Remark** We take an amortized causal discovery approach because it is hard to specify what assumptions a biological system ought to follow. This makes it difficult to provide identifiability guarantees without simplifying aspects of either the data or modeling framework. Instead, in Appendix C, we show that the differential network is a well-specified class of models. That is, there exist parametrizations of its attention architecture that map predicted graphs and global statistics (for hard and soft interventions) to the set of targets.

### 3.3. Implementation details

**Training procedure** The overall training loss is $\mathcal{L} = \mathcal{L}_G + \mathcal{L}_I$ with L2 weight regularization. Both $\mathcal{L}_G$ and $\mathcal{L}_I$ supervise the causal structure learner, while only the latter supervises the differential network. We use the synthetic and real datasets as follows.

1. Initialize the causal structure learner with pretrained weights from Wu et al. (2025) (except ablations).
2. Train both modules jointly on synthetic data.
3. Finetune both modules jointly with only $\mathcal{L}_I$ (no ground truth graphs), on each training split of the real datasets.

For the synthetic training data, we generated 8640 datasets

with hard and soft interventions, which vary in causal mechanism, intervention type, and graph topology. To emulate transcriptomics data, in which perturbation effect sizes are quantified in (log) fold change, we introduced soft interventions: "shift" $x \leftarrow f(\pi_x) + c$, where $c \in \mathbb{R}$, and "scale" $x \leftarrow cf(\pi_x)$, where $c$ is a positive scaling factor, with equal probability $c \lessgtr 1$. We also sampled interventions with multiple targets, to simulate off-target effects. Details are available in Section 4.2 and Appendix D.2.

**Model parameters** Marginal estimates $E'$ are computed using the classic FCI algorithm (Spirtes et al., 1995) with $k = 5$ and $T = 100$, and global statistic $\rho$ is pairwise correlation, as it is fast to compute and numerically stable. We swept over the number of differential network layers (Figure 4) on synthetic data, and we used 3 layers for $h_{\text{cat}}$ and 2 layers for $h_{\text{diff}}$. Following SEA, we adopted hidden dimension $d = 64$, the AdamW optimizer (Loshchilov & Hutter, 2019), learning rate 1e-4, batch size 16, and weight decay 1e-5. On the real data, where $N = 1000$, we changed to a batch size of 1, decreased the learning rate to 5e-6, and finetuned the models with half precision (FP16). We trained both $h_{\text{cat}}$ and $h_{\text{diff}}$ models on synthetic data, but only finetuned the $h_{\text{diff}}$ architecture on the real data (memory constraint).

## 4. Experiments

While our method is intended to be used in cases where true targets are unknown, we can only evaluate on datasets

where the target is known. Thus, we demonstrate that CDN can recover perturbation targets on seven transcriptomics datasets, with comparisons to state-of-the-art models for these applications (Section 4.1). For completeness, we also benchmarked CDN against multiple causal discovery algorithms for unknown interventions in variety of controlled settings (Section 4.2).

## 4.1. Biological experiments

**Datasets** We validate CDN on five Perturb-seq (Dixit et al., 2016) datasets (genetic perturbations) from Replogle et al. (2022) and Nadig et al. (2024); as well as two Sci-Plex (Srivatsan et al., 2020) datasets (chemical perturbations) from McFaline-Figueroa et al. (2024). Each dataset is a real-valued matrix of gene expression levels: the number of examples $M$ is the number of cells, the number of variables $N$ is the number of genes, and each entry is a log-normalized count of how many copies of gene $j$ was measured from cell $i$. In Perturb-seq datasets, we aim to recover the gene whose promoter was targeted by the CRISPR guide, and in Sci-Plex datasets, we aim to identify the gene that corresponds to the drug's intended target (discussion in Appendix D.1).

Biologically, many perturbations do not affect the cell's production of mRNA (they may affect other factors like protein activity, which transcriptomics does not measure). To ensure that $P_{\text{obs}}$ and $P_{\text{int}}$ are distinct, we filtered perturbations to those that induced over 10 differentially-expressed genes (statistically significant change, compared to control), of which the true target should be present. This excludes perturbations with insufficient cells (low statistical power), with minimal to no effect (uninteresting), and those that did not achieve the desired effect (low CRISPR efficiency). In the Sci-Plex data, several drugs have known off-targets, but we did not observe differential expression of any of them (adjusted p-value $\approx 1$).

**Evaluation** We consider two splits: seen and unseen cell lines. In the former, models may be trained on approximately half of the perturbations from each cell line, and are evaluated on the unseen perturbations. In the latter, we hold out one cell line at a time, and models may be trained on data from the remaining cell lines. To ensure that our train and test splits are sufficiently distinct, we cluster perturbations based on their log-fold change and assign each cluster to the same split (Figure 5). Since the Sci-Plex data are limited, we use them only as test sets without finetuning.

We limited the set of candidate targets to the top 1000 differentially expressed genes by log-fold change, per perturbation. If too few genes passed the significance threshold ($p < 0.05$), we added additional candidates with the largest log-fold change until we reached a minimum of 100 genes. Genetic perturbations have highly specific effects compared to chemical perturbations, so we focused on the subset for

which the intended target was *not* trivially identifiable as the gene with the largest log-fold change.[2]

While perturbation target prediction appears to be a simple classification task, standard metrics are not well-suited for these real data. Not all genes are present in the baselines' domain knowledge (e.g. isolated node in graph), so their effects as perturbations cannot be predicted. In addition, due to genetic redundancy, it is common for multiple perturbations to elicit similar responses (Kernfeld et al., 2023). An "incorrect" top 1 prediction may not necessarily reflect poor performance. Thus, we propose the following metrics.

- We order genes based on the similarity of their predicted effect to the ground truth target effect, or based on their predicted probability of being a target. The **rank** is the index of the true target, normalized by the number of candidate genes. Rank ranges from 0 (worst, bottom of list) to 1 (best, top of list).

- We measure the similarity between the perturbation effect (log-fold change) of the top 1 predicted target and the true target with the **Spearman rank correlation** $\rho$ and the **Pearson correlation** $r$. These range from -1 to 1, where 1 is best.

- To emulate "virtual screening," we plot **recall at p**: the fraction of targets recovered within the $p \in (0, 1)$ candidates. Recall ranges from 0 to 1, where 1 is best. Rank is equivalent to the expected recall at $p$.

**Baselines** We compare to both algorithms that infer the effects of perturbations on cells (used as scoring functions, Equation 3), and algorithms that directly predict targets. All baselines were run with their official implementations and/or latest releases. For details, please see Appendix B.2.

**GEARS** (Roohani et al., 2023) and **GENEPT** (Chen & Zou, 2023) predict the effects of unseen genetic perturbations. GEARS is a graph neural network that regresses log-fold change upon perturbation, using the Gene Ontology knowledge graph as an undirected backbone. GENEPT is a set of large language model-derived embeddings that have been shown to achieve state-of-the-art performance (Märtens et al., 2024) with simple downstream models like logistic regression (used here). Both are trained per cell line.

**PDGRAPHER** (Gonzalez et al., 2024) is a graph neural network that predicts the perturbation targets of genetic or chemical perturbations, on seen *or* unseen cell lines, with the Human Reference Interactome (Luck et al., 2020) as the knowledge graph. To the best of our knowledge, PDGRAPHER is the only published deep learning baseline for the target prediction task on transcriptomic data. Therefore, we compare against additional statistical and naive baselines.

---

[2]CDN predicts trivial perturbations nearly perfectly (Table 9).

**DGE** stands for differential gene expression (Love et al., 2014), i.e. the process of identifying statistically significant changes in genes between settings. Specifically, we run the Wilcoxon ranked-sum test with Benjamini-Hochberg correction (Wilcoxon, 1945; Benjamini & Hochberg, 2000) between control and perturbed cells, as implemented by Wolf et al. (2018). Genes are ranked by adjusted p-value. **LINEAR** and **MLP** take as input the mean expression of all perturbation targets, plus the top 2000 highly-variable genes (Wolf et al., 2018). They are trained to predict a binary label for each gene on each cell line independently.

**Results** CDN consistently outperforms all baselines on the five Perturb-seq datasets, in both the seen and unseen cell line settings (Table 1). There is minimal drop in between seen and unseen cell lines, suggesting that the functional characteristics of genetic perturbations are highly transferrable, even to different data distributions. Notably, no other deep learning method exceeds simple statistical tests (DGE). This is more evident in Figure 3, in which CDN achieves higher recall at $p$ at nearly all points. Finally, if we ablate the finetuning step ("synthetic" vs. "Perturb-seq"), we find that finetuning leads to significant improvements.

Chemical perturbations are much less specific than genetic perturbations (Figures 6 and 7), and it is less clear whether drug effects *should* be reflected in mRNA levels (via feedback mechanisms), as drugs act upon their protein products. Still, on the two chemical perturbation datasets, CDN finds the true target within the top 100 candidates for 3/6 cases (Table 2). Due to the low performance of DGE on several cases, we hypothesized that gene-level changes may not be as predictive. Indeed, an ablation of CDN, trained without node-level statistics, improves where DGE fails ("no $\mu$").

### 4.2. Synthetic experiments

While it would be ideal to evaluate all algorithms on real data, current causal discovery algorithms that support unknown interventions are not tractable on datasets with more than tens of variables. In fact, many baselines require hours on even $N = 10$ datasets, and they do not scale favorably (Table 6). Our transcriptomics datasets contain hundreds of genes, even after filtering to those that are differentially expressed (Figure 8). At the same time, existing models for biological perturbations all rely on some form of domain knowledge, and their performance is inseparable from the choice and quality of these external data. Synthetic data allow us to assess the model's capacity to predict intervention targets in isolation.

**Datasets** Each evaluation setting consists of 5 unseen, i.i.d. sampled graphs. To assess the generalization capacity of our framework, we tested the model on seen (linear) and unseen causal mechanisms (polynomial, sigmoid), and unseen soft interventions. To generate observational data, we

sampled Erdős-Rényi graphs with $N = 10, 20$ nodes and $E = N, 2N$ expected edges; causal mechanism parameters; and observations of each variable, in topological order. We sampled $3N$ distinct subsets of 1-3 nodes ($N$ each) as intervention targets. Hard interventions set $x \leftarrow z$, where $z$ is uniform. For soft interventions (Section 3.3), we trained on shift and tested on scale for synthetic experiments (see Table 10 for explanation). Real experiments trained on all interventions.

**Baselines** We compare against discrete and continuous causal discovery algorithms for unknown interventions. **UT-IGSP** (Squires et al., 2020) infers causal graphs and unknown targets by greedily selecting the permutation of variables that minimizes their proposed score function. **DCDI** (Brouillard et al., 2020) and **BACADI** (Hägele et al., 2023) are continuous causal discovery algorithms that fit generative models to the data, where the causal graph and intervention targets are model parameters. DCDI's -G and -DSF suffixes correspond to Gaussian and deep sigmoidal flow parametrizations of the likelihood. BACADI's -E and -M suffixes indicate empirical (standard) and mixture (bootstrap) variants.

**MB+CI** is inspired by Huang et al. (2020). We introduce a domain index (indicator of environment), compute its Markov boundary using graph lasso, and use conditional mutual information to identify true children of the domain index (intervention targets). **DCI** with stability selection (Belyaeva et al., 2021) predicts edge-level differences between two causal graphs. We take each node's proportion of changed edges as its likelihood of being an intervention target. While DCI was also motivated by biological applications, it only scales to around a hundred variables at most, so we evaluate it alongside other causal discovery methods here.

**Evaluation** In the synthetic case, we are not constrained by biological redundancy or incomplete predictions, so we report standard classification metrics: mean average precision (**mAP**) and area under the ROC curve (**AUC**). Both metrics are computed independently for each variable and averaged over all regimes of the same number of targets. Their values range from 0 to 1 (perfect). The mAP random baseline depends on the positive rate, while the AUC random baseline is 0.5 (per edge).

**Results** On synthetic data, CDN achieves high performance across intervention types and data-generating mechanisms (Table 3), while running in seconds (Table 6). As an ablation study, we investigated removing the graph loss and the pretrained weights, with no change to the graph-centric architecture. We found that supervising based on the graph is almost always helpful ("no $\mathcal{L}_G$" row), and pretraining leads to more stable training dynamics (Figure 10).

DCI and MB+CI are the best baselines, which is encourag-

Table 1: Results on 5 Perturb-seq datasets. Top: Train on all cell lines jointly. Bottom: Leave one cell line out of training. Test sets (unseen perturbations) are identical in both settings. CDN (synthetic) and DGE do *not* require training on real Perturb-seq data. Metrics, from left to right: normalized rank of ground truth, top 1 Spearman and Pearson correlations. Uncertainty quantification in Table 7. Runtimes in Table 11.

| Setting | Model | K562 (gw) rank | $\rho$ | $r$ | K562 (es) rank | $\rho$ | $r$ | RPE1 rank | $\rho$ | $r$ | HepG2 rank | $\rho$ | $r$ | Jurkat rank | $\rho$ | $r$ |
|---|---|---|---|---|---|---|---|---|---|---|---|---|---|---|---|---|
| Seen cell line | LINEAR | 0.51 | 0.18 | 0.23 | 0.48 | 0.22 | 0.25 | 0.49 | 0.42 | 0.52 | 0.51 | 0.35 | 0.43 | 0.50 | 0.20 | 0.23 |
| | MLP | 0.45 | 0.18 | 0.21 | 0.49 | 0.08 | 0.11 | 0.53 | 0.32 | 0.41 | 0.48 | 0.33 | 0.40 | 0.45 | 0.24 | 0.29 |
| | GENEPT | 0.52 | 0.29 | 0.32 | 0.42 | 0.16 | 0.19 | 0.41 | 0.37 | 0.46 | 0.31 | 0.32 | 0.40 | 0.51 | 0.31 | 0.36 |
| | GEARS | 0.56 | 0.19 | 0.22 | 0.45 | 0.18 | 0.20 | 0.49 | 0.35 | 0.44 | 0.54 | 0.32 | 0.40 | 0.50 | 0.22 | 0.26 |
| | PDG | 0.54 | 0.25 | 0.29 | 0.49 | 0.20 | 0.23 | 0.48 | 0.41 | 0.52 | 0.54 | 0.37 | 0.43 | 0.52 | 0.31 | 0.35 |
| | CDN (Perturb-seq) | **0.92** | **0.72** | **0.74** | **0.95** | **0.81** | **0.82** | **0.92** | **0.67** | **0.76** | **0.94** | **0.69** | **0.75** | **0.98** | **0.83** | **0.84** |
| Unseen cell line | DGE | 0.82 | 0.50 | 0.54 | 0.89 | 0.60 | 0.62 | 0.79 | 0.56 | 0.64 | 0.86 | 0.51 | 0.51 | 0.84 | 0.45 | 0.50 |
| | PDG | 0.39 | 0.10 | 0.13 | 0.44 | 0.24 | 0.28 | 0.54 | 0.38 | 0.48 | 0.48 | 0.34 | 0.41 | 0.57 | 0.31 | 0.35 |
| | CDN (synthetic) | 0.79 | 0.46 | 0.49 | 0.80 | 0.59 | 0.62 | 0.82 | 0.61 | 0.71 | 0.84 | 0.56 | 0.63 | 0.87 | 0.64 | 0.67 |
| | CDN (Perturb-seq) | **0.91** | **0.69** | **0.71** | **0.95** | **0.81** | **0.82** | **0.92** | **0.65** | **0.74** | **0.93** | **0.67** | **0.72** | **0.97** | **0.84** | **0.85** |

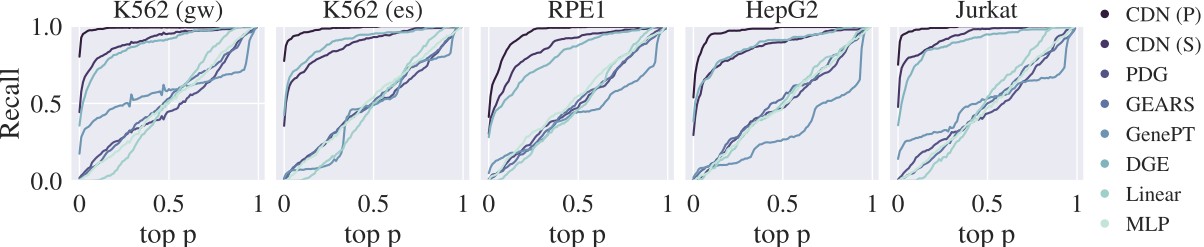

Figure 3: Recall at $p$ (normalized over number of candidate genes on 5 Perturb-seq datasets. For CDN, (P) and (S) denote "Perturb-seq" and "synthetic" training data.

Table 2: Rank of drug targets on chemical perturbation datasets (McFaline-Figueroa et al., 2024). A172 and T98G are *unseen* cell lines. Highlighted: Within top 100. Drug names and details in Appendix D.1.

| Model | A172 infig. | nint. | palb. | T98G doxo. | palb. | vola. |
|---|---|---|---|---|---|---|
| DGE | 0.12 | 0.64 | 0.28 | 0.94 | 0.24 | 0.83 |
| PDG | 0.72 | 0.35 | 0.39 | 0.06 | 0.34 | 0.42 |
| CDN | 0.14 | 0.81 | 0.01 | 0.89 | 0.93 | 0.53 |
| CDN (no $\mu$) | 0.88 | 0.44 | 0.58 | 0.06 | 0.54 | 0.65 |

ing, as they are designed with the intuition that detecting differences is easier than reproducing the entire causal graph. Surprisingly, most other baselines perform poorly at recovering intervention targets. In the case of DCDI and BACADI, this may be because it is hard to select a single sparsity threshold for varying sizes of intervention sets, and to balance the sparsity regularizer with the generative modeling objective.

## 5. Discussion

This work introduces causal differential networks (CDN), which focuses on the application of predicting perturbation targets for efficient experimental design. On a high level,

our goal is to lower the cost and scale of experiments, by reducing the search space. Our key technical contribution is the idea of using supervised causal discovery as a pretraining objective, to obtain dataset representations useful for downstream tasks like inferring targets. Specifically, CDN trains a *causal graph learner* that maps a pair of observational and interventional datasets to their data-generating mechanisms, and a *differential network* classifier to identify variables whose conditional independencies have changed. We demonstrated that our approach outperforms state-of-the-art methods in perturbation modeling and causal discovery, across seven transcriptomics datasets and a variety of synthetic settings. Overall, these results highlight the potential of our method to design more efficient experiments, by reducing the search space of perturbations.

There are several limitations of this work and directions for future research. First, our implementation does not address cyclic, feedback relationships or time-resolved, dynamic systems – both of which are common in biology. Modeling these cases may require data simulation schemes that can sample from the steady-state of a cyclic system, or architectures that directly predict the parameters of a system of stochastic differential equations (Lorch et al., 2024). In addition, current supervised causal discovery algorithms, which we use for the causal graph learner, tend to assume causal

Table 3: Intervention target prediction results on synthetic datasets with $N = 20$, $E = 40$. Top: hard interventions (uniform); bottom: soft interventions (scale). Neither polynomial nor "scale" soft interventions were seen during training. Number in parentheses indicates number of intervention targets. Uncertainty is standard deviation over 5 i.i.d. datasets. Runtimes in Table 6. Extended results in Table 12.

| Type | Model | Linear (1) | | Linear (3) | | Polynomial (1) | | Polynomial (3) | |
|---|---|---|---|---|---|---|---|---|---|
| | | mAP↑ | AUC↑ | mAP↑ | AUC↑ | mAP↑ | AUC↑ | mAP↑ | AUC↑ |
| Hard | UT-IGSP | $0.10_{\pm.01}$ | $0.63_{\pm.03}$ | $0.18_{\pm.01}$ | $0.52_{\pm.03}$ | $0.19_{\pm.02}$ | $0.70_{\pm.02}$ | $0.23_{\pm.03}$ | $0.56_{\pm.03}$ |
| | DCDI-G | $0.16_{\pm.02}$ | $0.50_{\pm.02}$ | $0.27_{\pm.05}$ | $0.50_{\pm.04}$ | $0.15_{\pm.03}$ | $0.45_{\pm.01}$ | $0.26_{\pm.02}$ | $0.51_{\pm.04}$ |
| | DCDI-DSF | $0.15_{\pm.02}$ | $0.46_{\pm.03}$ | $0.26_{\pm.03}$ | $0.47_{\pm.04}$ | $0.18_{\pm.05}$ | $0.48_{\pm.08}$ | $0.28_{\pm.04}$ | $0.52_{\pm.05}$ |
| | BACADI-E | $0.18_{\pm.09}$ | $0.71_{\pm.06}$ | $0.28_{\pm.02}$ | $0.69_{\pm.03}$ | $0.20_{\pm.06}$ | $0.84_{\pm.04}$ | $0.37_{\pm.07}$ | $0.82_{\pm.05}$ |
| | BACADI-M | $0.10_{\pm.02}$ | $0.66_{\pm.05}$ | $0.21_{\pm.01}$ | $0.63_{\pm.02}$ | $0.09_{\pm.02}$ | $0.71_{\pm.05}$ | $0.24_{\pm.03}$ | $0.70_{\pm.05}$ |
| | DCI | $0.58_{\pm.07}$ | $0.85_{\pm.04}$ | $0.63_{\pm.06}$ | $0.84_{\pm.03}$ | $0.55_{\pm.08}$ | $0.82_{\pm.04}$ | $0.56_{\pm.08}$ | $0.79_{\pm.05}$ |
| | MB+CI | $0.47_{\pm.12}$ | $0.68_{\pm.08}$ | $0.58_{\pm.09}$ | $0.71_{\pm.06}$ | $0.77_{\pm.05}$ | $0.88_{\pm.03}$ | $0.79_{\pm.04}$ | $0.88_{\pm.02}$ |
| | CDN | $\mathbf{0.82}_{\pm.05}$ | $\mathbf{0.96}_{\pm.02}$ | $\mathbf{0.88}_{\pm.04}$ | $\mathbf{0.95}_{\pm.02}$ | $\mathbf{0.85}_{\pm.05}$ | $\mathbf{0.97}_{\pm.01}$ | $\mathbf{0.90}_{\pm.04}$ | $\mathbf{0.96}_{\pm.02}$ |
| | CDN (no $\mathcal{L}_G$) | $0.65_{\pm.07}$ | $0.86_{\pm.06}$ | $0.72_{\pm.09}$ | $0.84_{\pm.07}$ | $0.81_{\pm.05}$ | $0.95_{\pm.02}$ | $0.86_{\pm.03}$ | $0.94_{\pm.01}$ |
| Soft | UT-IGSP | $0.10_{\pm.01}$ | $0.69_{\pm.03}$ | $0.19_{\pm.01}$ | $0.56_{\pm.03}$ | $0.18_{\pm.02}$ | $0.77_{\pm.02}$ | $0.22_{\pm.01}$ | $0.60_{\pm.01}$ |
| | DCDI-G | $0.17_{\pm.03}$ | $0.50_{\pm.05}$ | $0.23_{\pm.01}$ | $0.44_{\pm.03}$ | $0.14_{\pm.03}$ | $0.48_{\pm.03}$ | $0.30_{\pm.04}$ | $0.54_{\pm.04}$ |
| | DCDI-DSF | $0.17_{\pm.04}$ | $0.46_{\pm.04}$ | $0.24_{\pm.02}$ | $0.47_{\pm.03}$ | $0.16_{\pm.02}$ | $0.50_{\pm.05}$ | $0.26_{\pm.02}$ | $0.48_{\pm.02}$ |
| | BACADI-E | $0.25_{\pm.07}$ | $0.65_{\pm.03}$ | $0.40_{\pm.13}$ | $0.68_{\pm.08}$ | $0.43_{\pm.22}$ | $0.81_{\pm.09}$ | $0.49_{\pm.15}$ | $0.77_{\pm.05}$ |
| | BACADI-M | $0.11_{\pm.01}$ | $0.61_{\pm.02}$ | $0.26_{\pm.07}$ | $0.64_{\pm.06}$ | $0.25_{\pm.10}$ | $0.80_{\pm.09}$ | $0.38_{\pm.10}$ | $0.76_{\pm.06}$ |
| | DCI | $0.48_{\pm.02}$ | $0.78_{\pm.02}$ | $0.53_{\pm.06}$ | $0.77_{\pm.04}$ | $0.62_{\pm.09}$ | $0.89_{\pm.04}$ | $0.55_{\pm.04}$ | $0.80_{\pm.04}$ |
| | MB+CI | $0.46_{\pm.10}$ | $0.65_{\pm.06}$ | $0.49_{\pm.04}$ | $0.64_{\pm.03}$ | $\mathbf{0.97}_{\pm.02}$ | $0.98_{\pm.01}$ | $0.92_{\pm.05}$ | $0.94_{\pm.03}$ |
| | CDN | $\mathbf{0.83}_{\pm.10}$ | $\mathbf{0.95}_{\pm.05}$ | $0.76_{\pm.07}$ | $0.89_{\pm.04}$ | $\mathbf{0.97}_{\pm.03}$ | $\mathbf{1.00}_{\pm.00}$ | $\mathbf{0.96}_{\pm.02}$ | $\mathbf{0.99}_{\pm.01}$ |
| | CDN (no $\mathcal{L}_G$) | $0.77_{\pm.06}$ | $0.95_{\pm.03}$ | $\mathbf{0.81}_{\pm.08}$ | $\mathbf{0.93}_{\pm.04}$ | $0.94_{\pm.03}$ | $0.99_{\pm.00}$ | $0.96_{\pm.02}$ | $\mathbf{0.99}_{\pm.01}$ |
| | exclude scale | $0.32_{\pm.07}$ | $0.64_{\pm.04}$ | $0.34_{\pm.09}$ | $0.65_{\pm.08}$ | $0.06_{\pm.02}$ | $0.63_{\pm.02}$ | $0.08_{\pm.02}$ | $0.61_{\pm.06}$ |
| | exclude scale | $0.44_{\pm.04}$ | $0.92_{\pm.01}$ | $0.43_{\pm.09}$ | $0.90_{\pm.03}$ | $0.06_{\pm.01}$ | $0.72_{\pm.02}$ | $0.10_{\pm.02}$ | $0.73_{\pm.02}$ |

sufficiency, i.e. that there are no unseen confounders. This assumption is unrealistic in biology, as it is not physically possible to measure all relevant variables. Thus, there is an opportunity to extend this work for the latent confounding setting. Finally, since it is difficult to simulate biology, there will be an inevitable domain shift between synthetic and real data. However, there does exist plentiful unlabeled single-cell data (Consortium* et al., 2022), which could be used with self-supervised objectives to bridge the gap (Sun et al., 2020). In conclusion, we hope that this work will enable efficient experimental design and interpretation, as well as future machine learning efforts towards these tasks.

## Impact statement

Our work proposes a machine learning method to improve understanding of large-scale perturbation experiments, in the context of molecular biology. Towards broader impact, this work may facilitate better understanding of drug mechanisms; inform protocol design for cellular reprogramming; or identify causal targets of disease. While this work has numerous applications to the life sciences, it does not directly concern the design of molecules or other potentially harmful agents.

## Acknowledgements

This material is based upon work supported by the National Science Foundation Graduate Research Fellowship under Grant No. 1745302. We would like to acknowledge support from the NSF Expeditions grant (award 1918839: Collaborative Research: Understanding the World Through Code), Machine Learning for Pharmaceutical Discovery and Synthesis (MLPDS) consortium, and the Abdul Latif Jameel Clinic for Machine Learning in Health.

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

# A. Model architecture

## A.1. Attention layers

To balance expressivity and efficiency, our model is designed around series of "axial" attention layers, similar to Ho et al. (2019); Rao et al. (2021); Wu et al. (2025). The causal structure learner closely resembles the architecture of SEA (Wu et al., 2025), while the differential network is slightly different (since the input space is different). Briefly, the input to the causal structure learner is a matrix of shape $N \times N \times T \times d$. However, the (dense) global statistic does not span the $T$ dimension, and the local graph estimates are generally sparse. Thus, the causal graph learner separately attends over the $N \times N \times d$ and $K \times T \times d$ components ($K$ is the number of unique edges in sparse format), with "messages" to align the representations between each attention layer. In contrast, the differential network only operates over the dense $N \times (N + 1) \times d$ (or $2d$) representation. Thus, it omits the message passing layers and directly stacks dense attention layers.

## A.2. Edge embeddings

Since attention itself is invariant to input ordering, Transformer-style architectures use absolute (Devlin et al., 2019) or relative (Su et al., 2024) positional embeddings to distinguish input elements. Images and sequences have a natural ordering among their elements. This structure inherently induces distances between positions, which provides signal for learning models. In contrast, models over graphs should be equivariant to the labeling of nodes. That is, if $1 \rightarrow 2$ is an edge, while $2$ and $3$ are not connected, then re-labeling the nodes $(1, 2, 3) \mapsto (2, 3, 1)$ should result in edge $2 \rightarrow 3$ and no edge between $3$ and $1$.

If we assign an arbitrary ordering to each graphx, the Transformer will be able to distinguish between nodes. However, there are several practical issues. Since synthetic data are generated in the topological ordering of the nodes, always assigning the root nodes to low positions may lead to unintended information leakage. Furthermore, if the model is trained on graphs of up to $N = 100$, but tested on larger graphs, the positional embeddings associated with $N > 100$ will be entirely random and out of distribution. Following Wu et al. (2025), we randomly sample a permutation of the maximum graph size to each graph. Each edge embedding is the concatenation of its constituent nodes' embeddings, combined via a feed-forward network:

$$\text{Embed}(i, j) = \text{FFN}([\text{Embed}(i), \text{Embed}(j)]) \tag{9}$$

where $i, j$ are arbitrarily assigned, but consistent within the graph. This embedding is added to the hidden representation $h_{i,j}$ before the first layer of attention.

In the differential network, the additional "column" associated with node-level statistics is not subject to any embedding (which implicitly differentiates it from the remaining inputs), while the "row" identifies the node these statistics were computed for.

# B. Implementation details

## B.1. CDN training

We trained until convergence (no improvement for 50 epochs on synthetic; 10 epochs on Perturb-seq) and selected the best model based on validation mAP (Figure 4). The validation distribution was identical to training and did *not* contain test causal mechanisms. During training, we used 15 CPU workers (primarily for local graph estimates) and 1 A6000 GPU. CDN took around 6 hours to train on synthetic data, and 1 hour to finetune on real data. Unless otherwise noted, the causal structure learner was initialized from the SEA inverse covariance, FCI pretrained weights (Wu et al., 2025).

## B.2. Baselines

We used the latest releases of all baselines.

GEARS is a graph neural network that predicts the effects of unseen genetic perturbations, based on the Gene Ontology knowledge graph (Ashburner et al., 2000). We trained GEARS on each cell line separately, and predicted the effects of perturbing every expressed gene. Then we ranked candidates based on cosine similarity to the interventional data. Figure 9 depicts the low, but non-trivial proportion of differentially expressed genes for which GEARS was unable to make a prediction, due to lack of node coverage in their processed gene ontology graph (Ashburner et al., 2000). These genes were not considered in the rankings or evaluations for GEARS.

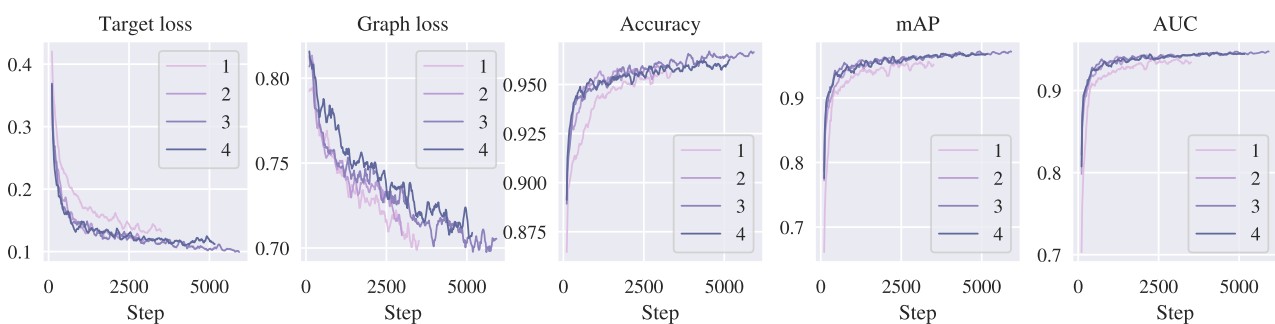

Figure 4: Hyperparameter search: number of differential network layers. We selected 3 layers based on validation metrics.

GENEPT utilized the v2 March 2024 update, which used newer models and additional protein data, compared to their initial paper (`GenePT_gene_protein_embedding_model_3_text`). We concatenated each perturbation's gene embedding to the log-fold change of the top 5000 highly-variable genes (Wolf et al., 2018) and trained a logistic regression model on each cell line to predict true vs. decoy perturbations. Candidates were ranked based on predicted probability. Of the 2,842 unique genetic perturbations, only 2,819 (99.1%) mapped to GENEPT embeddings. The remainder used the mean gene embedding (within the dataset) as the language-based embedding.

PDGRAPHER was published on the union of three distinct knowledge graphs, but their harmonized graphs were not available publicly, and the authors did not respond to requests for data sharing. As a result, we relied on solely the human reference interactome (Luck et al., 2020) graph, as it was the only one of the three that could be easily processed. All test perturbation targets could be inferred through PDGRAPHER.

UT-IGSP used the official implementation from commit `7349396`, since the repository is not back-compatible.

BACADI is evaluated using the fully-connected implementation, since it performed better than the linear version. However, since the linear version is significantly faster, we include its runtime in Table 6.

MB+CI was implemented using scikit-learn (Pedregosa et al., 2011).

## C. Theoretical motivation

While this paper focuses on biological applications, we show that the differential network is well-specified as a class of models, i.e. a correct solution is contained within the space of possible parametrizations.

**Preliminary note**   In this analysis, we assume that the "causal" representations $h$ contains enough information both to retain global statistics and recover the true graph $E$. The former is reasonable due to high model capacity, and the latter is based on high empirical performance in graph reconstruction (Wu et al., 2025). We emphasize that the latter is an empirical judgment, which may not hold on all datasets in practice.

**Attention-based architecture**   Our differential network is implemented using an attention layer, which is composed of two self-attention layers (one along each axis of the adjacency matrix) and a feed-forward network. For simplicity, we follow prior work (Yun et al., 2019) and ignore layer normalization and dropout.

Our inputs $h \in \mathbb{R}^{2d \times N \times N}$ are $2d$-dimension features, which represent a pair of $N \times N$ causal graphs. We use $h_{\cdot,j}$ to denote a length $N$ row for a fixed column $j$, and $h_{i,\cdot}$ to denote a length $N$ column for a fixed row $i$. The attention layer implements:

$$\text{Attn}_{\text{row}}(h_{\cdot,j}) = h_{\cdot,j} + W_O W_V h_{\cdot,j} \cdot \sigma \left[ (W_K h_{\cdot,j})^T W_Q h_{\cdot,j} \right],$$
$$\text{Attn}_{\text{col}}(h_{i,\cdot}) = h_{i,\cdot} + W_O W_V h_{i,\cdot} \cdot \sigma \left[ (W_K h_{i,\cdot})^T W_Q h_{i,\cdot} \right],$$
$$\text{FFN}(h) = h + W_2 \cdot \text{ReLU}(W_1 \cdot h + b_1) + b_2,$$

where $W_O \in \mathbb{R}^{2d \times 2d}, W_V, W_K, W_Q \in \mathbb{R}^{2d \times 2d}, W_2 \in \mathbb{R}^{2d \times m}, W_1 \in \mathbb{R}^{m \times 2d}, b_2 \in \mathbb{R}^{2d}, b_1 \in \mathbb{R}^m$, and $m$ is the FFN hidden dimension. We have omitted the $i$ and $j$ subscripts on the $W$s, but they use separate parameters. Any self attention can take on the identity mapping by setting $W_O, W_V, W_K, W_Q$ to $2d \times 2d$ matrices of zeros.

**Hard interventions**  Let $G = (V, E)$ be a causal graphical model associated with data distribution $P_X$. Let $G' = (V, E')$ and $\tilde{P}_X$ denote the causal graph and data distribution after an unknown intervention, with ground truth targets $I \subsetneq V$. For convenience, we use $E, E'$ both to denote sets of edges, as well as the equivalent adjacency matrices.

In the case of perfect interventions,

$$f(x_i) \leftarrow z_i, \forall i \in I \tag{10}$$

where $z_i \perp\!\!\!\perp X$ are independent random variables. $\tilde{P}_X$ is associated with mutilated graph $E'$, where

$$E' = E \setminus \bigcup_{i \in I} \{(j, i)\}_{(j,i) \in E}. \tag{11}$$

In terms of the associated adjacency matrices, $E - E'$ has 1s in each column $i \in I$ and 0s elsewhere.

Here, the attention layer should implement $h - h'$, so that when we collapse over the incoming edges, the output is non-zero only at the edge differences. Suppose the first dimension of the $2d$ feature stores $E$, and the second dimension stores $E'$. The row self-attention implements the identity (in the first two dimensions). Then we can set $W_{K,Q}$ to zero, $W_V$ to the identity, and $W_O$ to

$$W_O = \begin{bmatrix} 1 & -1 \\ 0 & 0 \end{bmatrix} - \begin{bmatrix} 1 & 0 \\ 0 & 1 \end{bmatrix} \tag{12}$$

to account for the residual. The FFN implements the identity, so that when we take the mean over along the rows, we recover non-zero elements at all nodes whose incoming edges were removed.

**Soft interventions**  We also study soft interventions the context of causal models with non-multiplicative noise, in which intervention targets are scaled by constant factors,

$$f(x_i) \leftarrow c_i f(x), c_i > 0 \tag{13}$$

where $c_i$ are sampled at random per synthetic dataset. This choice is inspired by the fact that biological perturbation effects are measured in fold-change. Here, the adjacency matrices are the same, but global statistics differ. In particular, we focus on two statistics: the correlation matrix $R$ and the covariance matrix $\Sigma$. Note that while we do not explicitly provide the covariance matrix as input, we do provide marginal variances, from which the same information can be derived.

Suppose $x$ is an intervention target.

- $R - R'$ is non-zero in all entries $i, j$ and $j, i$ where $i$ is a descendent of $x$, and $j$ is any node for which $R_{i,j} \neq 0$ (e.g. ancestors, descendants, and $x$, if $P_X$ is faithful to $G$).

- $\Sigma - \Sigma'$ is non-zero in all entries $i, j$ and $j, i$ where $i$ is a descendent of $x$ or $i = x$, and $j$ is any node for which $\Sigma_{i,j} \neq 0$ (e.g. ancestors, descendants, and $x$, if $P_X$ is faithful to $G$).

All descendants are always affected, due to the non-multiplicative noise term. These two differ in the row and column that correspond to $x$ since

$$\text{Corr}(c \cdot x, y) = \text{Corr}(x, y) \tag{14}$$
$$\text{Cov}(c \cdot x, y) = c \cdot \text{Cov}(x, y). \tag{15}$$

Therefore, to identify $x$, we should find the index in which $\Sigma$ differs but not $R$. Suppose that dimensions 3-6 of $h$ encode $R, R', \Sigma, \Sigma'$. Following the same strategy as the hard interventions, we can use the row attention to compute $R - R', \Sigma - \Sigma'$ and store them in dimensions 3, 4. Then we use the column attention to filter out variables that are independent from $x$ by storing the sum of each column in dimensions 5, 6. While not strictly impossible, it is unlikely that a variable dependent on $x$ would result in a column that sums to exactly 0. Thus, all columns with non-zero sums are either ancestors, descendants, or $x$. The feedforward network implements

$$\text{FFN}(h_{\cdot, 3-6}) = \begin{cases} 1 & h_{\cdot,3} = 0, h_{\cdot,4} \neq 0, h_{\cdot,5} \neq 0 \\ 0 & \text{otherwise.} \end{cases} \tag{16}$$

This results in 1s in the rows and columns where $\Delta R$ and $\Delta \Sigma$ differ. After collapsing over incoming edges and normalizing to probabilities, the maximum probabilities can be found at the intervention targets.

**Supporting both intervention types**  Recall that the final output layer is a linear projection from $2d$ to 1. If this layer implements a simple summation over all $2d$, the predicted intervention targets are consistent with both hard and soft interventions. For soft interventions, $E = E'$, so the hard intervention dimensions will be 0. Likewise, for hard interventions, both $R$ and $\Sigma$ will differ as the same locations, as the underlying variable has changed, so the soft intervention dimensions will be 0. Since the two techniques produce mutually exclusive predictions, this means that both hard and soft interventions can co-exist and be detected on different nodes.

# D. Datasets

### D.1. Biological datasets

**Data processing**  We converted all single cell datasets to log-normalized, transcripts per 10,000 UMIs. Perturb-seq genes were mapped to standard identifiers and filtered by the authors. Sci-Plex dataset variables represented genes that appeared in at least 5,000 cells (threshold chosen to achieve a similar number of genes). We performed differential expression analysis via the `scanpy` package (Wolf et al., 2018), using the Wilcoxon signed-rank test (Wilcoxon, 1945) with Benjamini-Hochberg p-value correction and a threshold of adjusted p-value $< 0.05$. Table 4 reports statistics of the raw, unprocessed datasets.

For Perturb-seq datasets, we kept perturbations with $> 10$ differentially-expressed genes (DEGs), and clustered them using k-means with $k = 200$, chosen heuristically based on log-fold change heatmaps (Figure 5). These clusters were used to inform data splits for seen cell lines, where the largest cluster was allocated to the training set, and all remaining clusters were split equally among train and test. The largest cluster(s) appear to contain perturbations with smaller effects. For Sci-Plex datasets, only 6 drug perturbations across 2 cell lines resulted in differential expression of their known protein targets (Supplementary Table 8 from McFaline-Figueroa et al. (2024)). Therefore, we used these exclusively as test sets. Table 5 reports statistics of the final, processed datasets. Figure 8 plots the full distribution of number of cells and DEGs per perturbation.

**Top differentially expressed genes**  Here, we limited our analysis to the top 1000 DEGs. In the literature, it is quite common to restrict analysis to subsets of genes, though the threshold and criteria may vary (Luecken & Theis, 2019). Selecting the top 1000 genes (based on differential expression p-value) is reasonably permissive, since these genes are selected per perturbation. For a sense of scale, Nadig et al. (2024) writes:

> "In a genome-scale Perturb-seq screen, we find that a typical gene perturbation affects an estimated 45 genes, whereas a typical essential gene perturbation affects over 500 genes."

A classic approach for identifying drug targets argues that targets are not highly differentially expressed, but network-based approaches are useful for predicting them from gene expression (Isik et al., 2015).This work also samples 1000 candidate targets as decoys.

**Gene co-expression**  Figures 6 and 7 depict the differences in gene co-expression matrices between control and perturbed cells. Genetic perturbations tend to have much clearer phenotypes compared to chemical perturbations, whose effects are more diffuse.

**Sci-Plex drug targets**  The full names of the Sci-Plex drugs and their targets are as follows. These targets were collated by the authors (McFaline-Figueroa et al., 2024), and are each are well-documented in the literature.

- Infigratinib ("infig") binds target FGFR1 with nano-molar affinities (Knox et al., 2024).

- Nintedanib ("nint") and target FGFR1 have been probed through structural studies (Hilberg et al., 2008).

- Palbociclib ("palb") has been co-crystallized (PDB 5L2I) with its target CDK6.

- Doxorubicin ("doxo") and target TOP2A are associated in a number of works (Knox et al., 2024).

- Volasertib ("vola") has been co-crystallized (PDB 3FC2) with PLK1 and a number of works confirm specificity for PLK2 (Chen et al., 2021).

Table 4: Extended biological dataset statistics (raw).

| Type | Source | Accession | Cell line | # Perts | # Genes | # NTCs | # Cells |
|---|---|---|---|---|---|---|---|
| Genetic | Replogle et al. (2022) | Figshare 20029387 | K562 gw | 9,866 | 8,248 | 75,328 | 1,989,578 |
| | | | K562 es | 2,057 | 8,563 | 10,691 | 310,385 |
| | | | RPE1 | 2,393 | 8,749 | 11,485 | 247,914 |
| | Nadig et al. (2024) | GSE220095 | HepG2 | 2,393 | 9,624 | 4,976 | 145,473 |
| | | | Jurkat | 2,393 | 8,882 | 12,013 | 262,956 |
| Chemical | McFaline-Figueroa et al. (2024) | GSM7056151 | A172 | 23 | 8,393 | 8,660 | 58,347 |
| | | | T98G | 23 | 8,393 | 6,921 | 58,347 |

Table 5: Extended biological dataset statistics (processed).

| | Perturbations | | | | Genes | | Cells |
|---|---|---|---|---|---|---|---|
| Dataset | Train | Test | Trivial | Non-trivial | Unique # DE | Median # DE | |
| K562 gw | 1089 | 678 | 587 | 91 | 7,378 | 81 | 492,096 |
| K562 es | 640 | 420 | 348 | 72 | 8,492 | 226 | 213,552 |
| RPE1 | 564 | 397 | 233 | 164 | 8,641 | 399 | 179,696 |
| HepG2 | 364 | 263 | 162 | 101 | 9,282 | 271 | 79,309 |
| Jurkat | 679 | 333 | 262 | 71 | 8,432 | 162 | 174,698 |
| A172 | — | 3 | — | — | 445 | 324 | 18,196 |
| T98G | — | 3 | — | — | 1,644 | 508 | 13,126 |

It is important to note that these drugs are also associated with known off-targets. However, we did not observe differential expression of any off-target, adjusted $p \approx 1$, perhaps because the impact on their mRNA expression is negligible. Protein activity would be more suitable for quantifying the impact of a drug on a cell. However, this cannot be measured at scale. In fact, even protein abundance is difficult to measure at single-cell resolution, as it relies on mass spectrometry to differentiate proteins (Newman et al., 2006), while mRNA can be exhaustively enumerated and sequenced (based on the known human genome). Therefore, this chemical perturbation study is primarily a proof of concept, to probe whether targets can be inferred from mRNA expression, and if so, to what extent.

### D.2. Synthetic data

Synthetic data were generated using code modified from DCDI (Brouillard et al., 2020) for soft interventions. We used the following implementations for causal mechanisms $f(x)$, where $x$ is the variable in question, $M$ is a binary mask for the parents of $x$, $X$ contains measurements of all variables, $E$ is independent Gaussian noise, and $W$ is a random weight matrix.

- Linear: $f(x) = MXW + E$.

- Neural network, non-additive $f(x) = \text{Tanh}((X, E)W_{\text{in}})W_{\text{out}}$

- Neural network, additive: $f(x) = \text{Tanh}(XW_{\text{in}})W_{\text{out}} + E$

- Polynomial: $f(x) = W_0 + MXW_1 + MX^2W_2 + E$

- Sigmoid: $f(x) = \sum_{i=1}^{d} W_i \cdot \text{sigmoid}(X_i) + E$

Linear and the neural network variants were used for training. Polynomial and sigmoid were only used for testing.

Root causal mechanisms are uniform. For hard interventions, we set $f(x) \leftarrow z$, where $z \sim \text{Uniform}(-1, 1)$. For soft "scale" interventions, we set

$$f(x) \leftarrow z_1^{\text{Sign}(z)} \cdot f(x)$$
$$z_1 \sim \text{Uniform}(2, 4)$$
$$z \sim \text{Uniform}(-1, 1).$$

That is, we multiply $f(x)$ by a scaling factor that is equal probability $\lessgtr 1$ (constant across all observations).

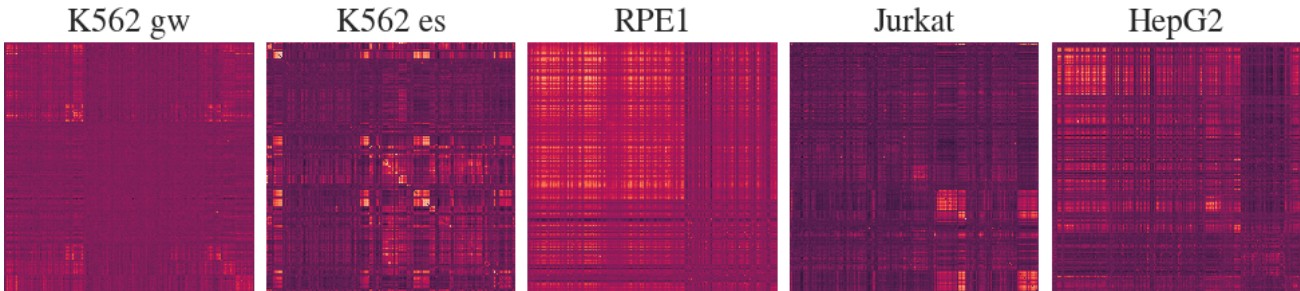

Figure 5: Heatmap of correlation between log-fold change, sorted by cluster, used for data splits.

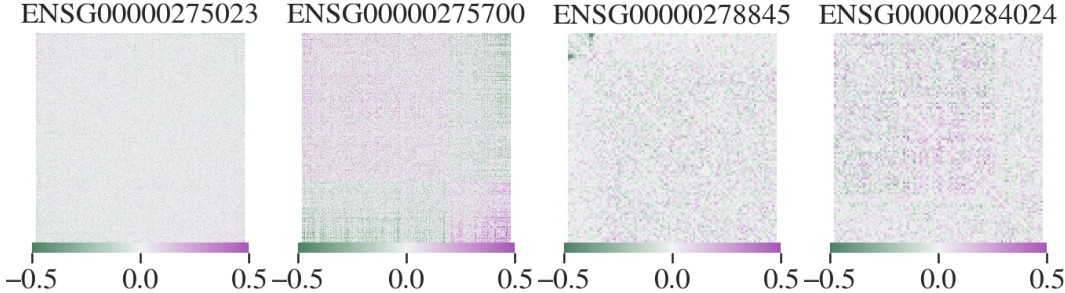

Figure 6: Difference between observational and interventional correlation matrices on K562 genome-wide. In some cases, the changes are minimal and specific; in others, they are more diffuse.

For soft "shift" interventions, we set

$$f(x) \leftarrow f(x) + \text{Sign}(z) \cdot z_1$$
$$z_1 \sim \text{Uniform}(2, 4)$$
$$z \sim \text{Uniform}(-1, 1).$$

That is, we translate $f(x)$ by a scaling factor that is equal probability $\lessgtr 0$ (constant across all observations).

## E. Additional analyses

**Runtime** We compared the runtimes of various algorithms on the Perturb-seq and synthetic datasets. On the Perturb-seq data, all models finished running within minutes with the exception of GEARS (Figure 11). Runtimes of the various causal algorithms varied significantly (Table 6). The slowest method was DCDI, with an average runtime of around 10 hours on $N = 20$ datasets, while the fastest were UT-IGSP and the MLP variant of CDN. All models were benchmarked on equivalent hardware (A6000 GPU, 1 CPU core).

**Perturb-seq uncertainty quantification** Due to space limitations, we report uncertainty estimates in Tables 7 and 8. Multiple baselines produce deterministic results (LINEAR, DGE, GENEPT), so instead of model randomness, we report uncertainties over the sampling of single cells (Table 7). Specifically, for each perturbation with $M$ cells, we sample

$$M' = \min(M, \max(50, 0.8M)) \tag{17}$$

cells uniformly at random, repeated 5 times. We also consider the robustness of our model to samplings of the test set (Table 8). Typically, the largest cluster (assigned to train, e.g. nearly 1/3 of K562) contains weak perturbations with minimal phenotype, so different clusterings don't introduce significant heterogeneity. Instead, we can partition all perturbations uniformly into 5 sets for the unseen cell line setting, across which the variation in results is minimal.

**Perturb-seq trivial perturbations** For the main results in Table 1, we focus on genetic perturbations for which the intended gene target was *not* the gene with the greatest log-fold change. This is because genetic perturbations are (unrealistically) specific, compared to drugs or transcription factors. In Table 9, we show that our model performs nearly perfectly in this easy setting.

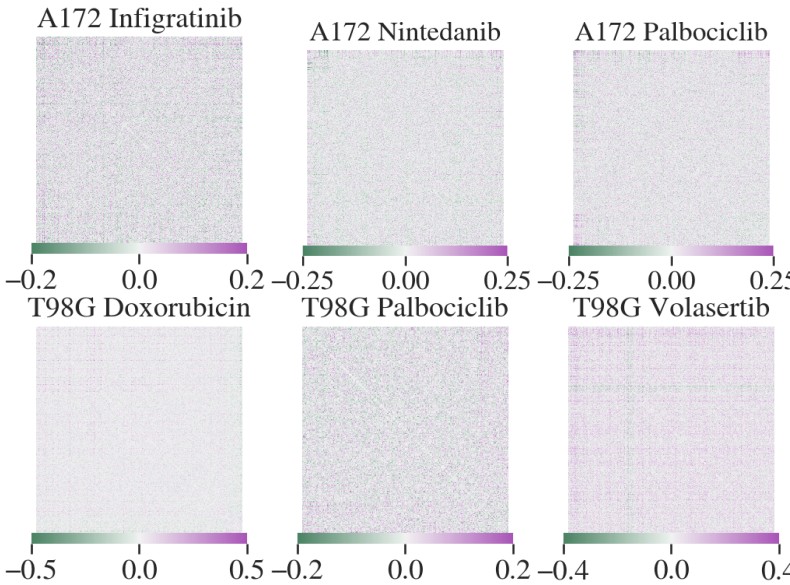

Figure 7: Difference between observational and interventional correlation matrices on Sci-Plex datasets. Effects are much more diffuse than in genetic perturbations (Figure 6).

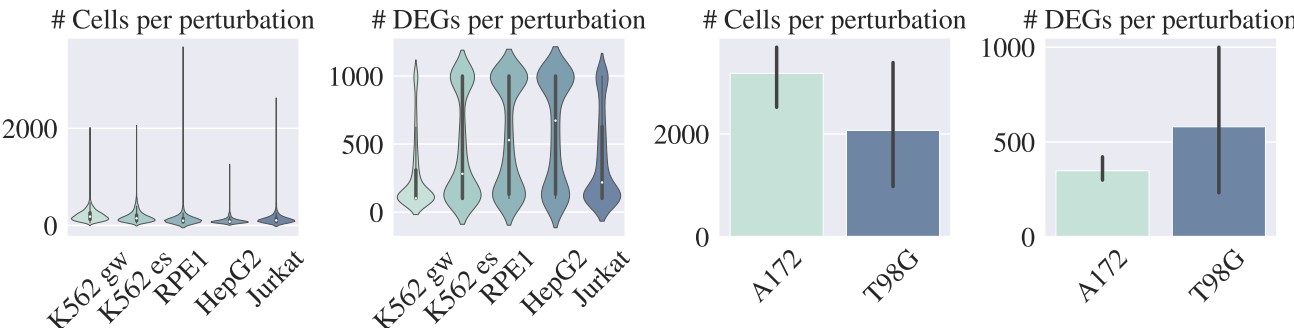

Figure 8: Transcriptomics dataset statistics, after processing. See Table 5 for details.

**Graph loss**  We assess the value of predicting the causal graph, as opposed to only training a supervised classifier (Figure 10). Without modifying the architecture (which has inductive biases towards the graph modality), we train a version of CDN without any graph labels (causal graph learner no longer pretrained; remove graph loss $\mathcal{L}_G$). Across the validation loss and various metrics, we find that the full CDN is more stable, and leads to better results. This model corresponds to the "no $\mathcal{L}_G$" test results in Table 3.

**Additional synthetic results**  During model development, we realized that if the model is allowed to train on all types of perturbations, it can easily overfit and learn these distributional shifts very well. For example, in Table 10, the models trained on both types of soft interventions performs perfectly (unrealistic) in many cases, even though the graphs and causal mechanism parameters are completely novel. Therefore, to assess whether our model can generalize, we chose to hold out a class of soft intervention for our main results in Table 3.

Table 11 reports results on larger graphs. There is some drop in performance, especially Polynomial. This could be attributed to error propagation from the causal graph predictor, which performs less well (at predicting graphs) on larger graphs, and could potentially be ameliorated by finetuning on larger synthetic data (similar to how Perturb-seq models were finetuned to be larger). Linear + Soft remains quite good, perhaps because the theory (Appendix C) suggests that the summary statistics are sufficient for predicting targets, without the graph prediction.

Table 12 reports results on all synthetic test datasets, averaging over all interventions for each dataset. We also include the

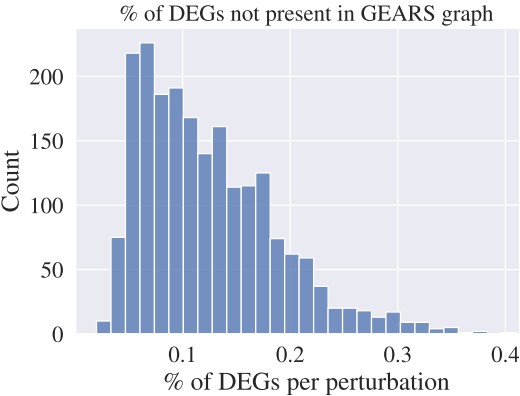

Figure 9: Percentage of differentially expression genes (per perturbation) that were *not* in the GEARS knowledge graph, and whose effects could not be predicted.

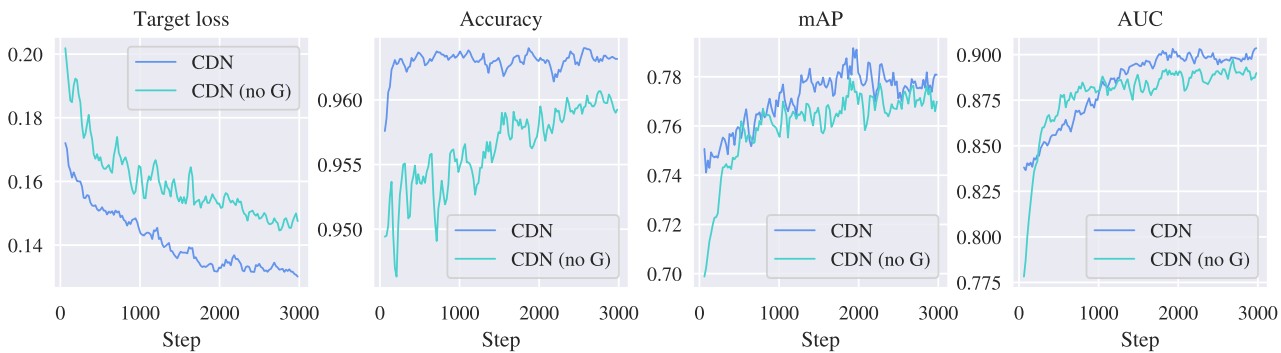

Figure 10: Ablation study: We compare CDN trained on synthetic data, to the same model without any ground truth graph information (no pretraining or $\mathcal{L}_G$). We find that graph information leads to more stable and better results. Validation metrics and loss. One step is one batch (16 samples).

"full" version of CDN, i.e. trained on all mechanisms, in this analysis.

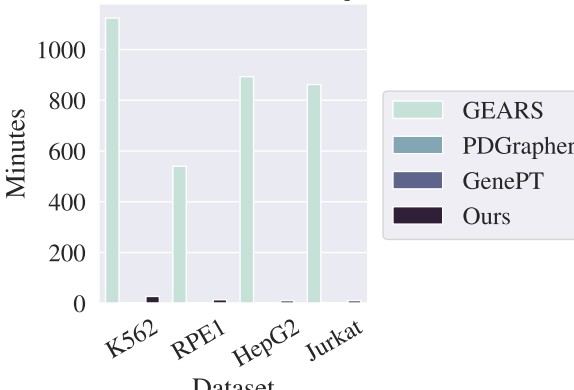

Inference runtime on Perturb-seq cell lines

GEARS
PDGrapher
GenePT
Ours

Figure 11: Inference runtimes on Perturb-seq datasets. K562 datasets are reported together. All models run on a single A6000 GPU, no constraint on memory (up to 500G). Only GEARS required over ~20G of memory. To the best of our ability, we normalized batch size to 1.

Table 6: Runtimes on synthetic datasets (sec).

| Nodes | Model | Min | Max | Mean | Std |
|---|---|---|---|---|---|
| 10 | UT-IGSP | 1 | 98 | 26 | 31 |
| | DCDI-G | 31 | 27908 | 18731 | 9857 |
| | DCDI-DSF | 7118 | 44440 | 23798 | 5688 |
| | DCI | 41 | 1438 | 404 | 365 |
| | BACADI | 4005 | 9483 | 6284 | 1584 |
| | BACADI-L | 1076 | 1403 | 1272 | 76 |
| | CDN (MLP) | 1 | 5 | 1 | 1 |
| | CDN | 17 | 147 | 39 | 27 |
| 20 | UT-IGSP | 1 | 78 | 19 | 26 |
| | DCDI-G | 45 | 46281 | 35032 | 16068 |
| | DCDI-DSF | 23181 | 55406 | 30076 | 7200 |
| | DCI | 325 | 21414 | 4415 | 4707 |
| | BACADI | 15082 | 50737 | 26020 | 9044 |
| | BACADI-L | 2862 | 3819 | 3461 | 252 |
| | CDN (MLP) | 1 | 3 | 1 | 0 |
| | CDN | 36 | 86 | 54 | 13 |

Table 7: Uncertainty quantification on Perturb-seq datasets, seen cell lines, by sub-sampling to $80\%$ of cells per perturbation (or 50, whichever is higher). Note that results may be slightly different from Table 1 due to sub-sampling. Standard deviation reported over 5 seeds. GENEPT performance is highly variable. DGE is quite sensitive to sub-sampling on K562 genome-wide.

| Model | K562 gw | | | K562 es | | | RPE1 | | |
|---|---|---|---|---|---|---|---|---|---|
| | rank | $\rho$ | $r$ | rank | $\rho$ | $r$ | rank | $\rho$ | $r$ |
| LINEAR | 0.50±.00 | 0.17±.01 | 0.20±.01 | 0.50±.01 | 0.25±.01 | 0.28±.01 | 0.48±.00 | 0.40±.01 | 0.49±.01 |
| MLP | 0.48±.00 | 0.17±.00 | 0.20±.00 | 0.48±.00 | 0.16±.00 | 0.19±.00 | 0.53±.00 | 0.30±.00 | 0.39±.00 |
| DGE | 0.72±.01 | 0.48±.02 | 0.52±.02 | 0.86±.00 | 0.59±.02 | 0.61±.02 | 0.74±.00 | 0.56±.00 | 0.64±.00 |
| GENEPT | 0.54±.09 | 0.26±.08 | 0.29±.08 | 0.50±.08 | 0.36±.10 | 0.39±.10 | 0.51±.10 | 0.43±.08 | 0.51±.07 |
| GEARS | 0.50±.01 | 0.19±.01 | 0.22±.01 | 0.51±.00 | 0.20±.01 | 0.23±.00 | 0.46±.00 | 0.33±.01 | 0.41±.02 |
| PDG | 0.49±.01 | 0.22±.00 | 0.25±.00 | 0.50±.01 | 0.28±.01 | 0.32±.01 | 0.49±.01 | 0.38±.01 | 0.47±.01 |
| CDN | **0.91**±.01 | **0.71**±.02 | **0.72**±.02 | **0.95**±.00 | **0.80**±.03 | **0.81**±.03 | **0.92**±.00 | **0.65**±.02 | **0.74**±.01 |

| Model | HepG2 | | | Jurkat | | |
|---|---|---|---|---|---|---|
| | rank | $\rho$ | $r$ | rank | $\rho$ | $r$ |
| LINEAR | 0.49±.00 | 0.34±.01 | 0.42±.01 | 0.50±.00 | 0.23±.01 | 0.27±.01 |
| MLP | 0.50±.00 | 0.34±.00 | 0.41±.00 | 0.47±.00 | 0.25±.00 | 0.31±.00 |
| DGE | 0.83±.00 | 0.49±.01 | 0.50±.01 | 0.78±.01 | 0.44±.02 | 0.49±.02 |
| GENEPT | 0.50±.13 | 0.39±.07 | 0.45±.06 | 0.51±.07 | 0.36±.11 | 0.40±.11 |
| GEARS | 0.51±.00 | 0.38±.00 | 0.45±.00 | 0.52±.01 | 0.24±.00 | 0.30±.00 |
| PDG | 0.49±.00 | 0.34±.01 | 0.41±.01 | 0.49±.01 | 0.29±.00 | 0.34±.00 |
| CDN | **0.93**±.00 | **0.67**±.01 | **0.73**±.01 | **0.96**±.01 | **0.83**±.04 | **0.85**±.03 |

Table 8: To quantify the uncertainty across draws of the test set (perturbations, instead of cells), we partition all perturbations uniformly into 5 sets for the unseen cell line setting on K562 genome-wide. The variation across test sets is minimal.

| Test Split | rank | $\rho$ | $r$ | Recall@1 | Recall@5 | Recall@20 |
|---|---|---|---|---|---|---|
| Original ($n = 678$) | 0.974 | 0.915 | 0.921 | 0.699 | 0.900 | 0.957 |
| New splits ($n = 353, \times 5$) | 0.973 ±0.009 | 0.917 ±0.017 | 0.923 ±0.016 | 0.696 ±0.031 | 0.908 ±0.032 | 0.961 ±0.021 |

Table 9: Results on 5 Perturb-seq datasets, *trivial* perturbations (intended target has largest log-fold change, though not necessarily smallest adjusted p-value). Top: Train on all cell lines jointly. Bottom: Leave one cell line out of training. Test sets (unseen perturbations) are identical in both settings. CDN (synthetic) and DGE do *not* require training on real Perturb-seq data. Metrics, from left to right: normalized rank of ground truth, top 1 Spearman and Pearson correlations.

| Setting | Model | K562 (gw) | | | K562 (es) | | | RPE1 | | | HepG2 | | | Jurkat | | |
|---|---|---|---|---|---|---|---|---|---|---|---|---|---|---|---|---|
| | | rank | $\rho$ | $r$ | rank | $\rho$ | $r$ | rank | $\rho$ | $r$ | rank | $\rho$ | $r$ | rank | $\rho$ | $r$ |
| | LINEAR | 0.49 | 0.18 | 0.21 | 0.50 | 0.28 | 0.31 | 0.48 | 0.37 | 0.45 | 0.48 | 0.34 | 0.43 | 0.48 | 0.24 | 0.29 |
| | MLP | 0.48 | 0.16 | 0.20 | 0.47 | 0.17 | 0.21 | 0.54 | 0.29 | 0.37 | 0.51 | 0.35 | 0.42 | 0.48 | 0.26 | 0.32 |
| Seen | GENEPT | 0.60 | 0.36 | 0.39 | 0.47 | 0.23 | 0.26 | 0.57 | 0.49 | 0.55 | 0.44 | 0.35 | 0.41 | 0.55 | 0.48 | 0.53 |
| cell line | GEARS | 0.49 | 0.15 | 0.18 | 0.51 | 0.25 | 0.27 | 0.49 | 0.32 | 0.40 | 0.48 | 0.32 | 0.40 | 0.47 | 0.20 | 0.24 |
| | PDG | 0.49 | 0.23 | 0.26 | 0.52 | 0.31 | 0.34 | 0.51 | 0.34 | 0.42 | 0.46 | 0.34 | 0.40 | 0.49 | 0.30 | 0.35 |
| | CDN (Perturb-seq) | **0.99** | **0.97** | **0.97** | **0.99** | **0.98** | **0.98** | **0.99** | **0.93** | **0.94** | **0.99** | **0.95** | **0.96** | **0.99** | **0.96** | **0.96** |
| | DGE | 0.87 | 0.68 | 0.70 | 0.92 | 0.77 | 0.79 | 0.85 | 0.73 | 0.77 | 0.89 | 0.59 | 0.60 | 0.91 | 0.73 | 0.77 |
| Unseen | PDG | 0.40 | 0.12 | 0.15 | 0.41 | 0.22 | 0.24 | 0.48 | 0.28 | 0.37 | 0.47 | 0.33 | 0.41 | 0.53 | 0.28 | 0.33 |
| cell line | CDN (synthetic) | 0.91 | 0.72 | 0.74 | 0.92 | 0.83 | 0.84 | 0.94 | 0.83 | 0.86 | 0.90 | 0.73 | 0.77 | 0.95 | 0.82 | 0.84 |
| | CDN (Perturb-seq) | **0.98** | **0.95** | **0.95** | **0.99** | **0.95** | **0.96** | **0.99** | **0.92** | **0.94** | **0.99** | **0.95** | **0.96** | **0.99** | **0.96** | **0.96** |

Table 10: Intervention target prediction results on synthetic datasets with $N = 20, E = 40$. Top: hard interventions (uniform); bottom: soft interventions (scale). We found that if the model is allowed to see the class of soft intervention ("all mechanisms") during training, it can easily learn (overfit) the distribution shift perfectly, even though test datasets and graphs are novel. Thus, our main results in Table 3 treat "scale" as an unseen soft intervention.

| Type | Setting | Linear (1) | | Linear (3) | | Polynomial (1) | | Polynomial (3) | |
|---|---|---|---|---|---|---|---|---|---|
| | | mAP↑ | AUC↑ | mAP↑ | AUC↑ | mAP↑ | AUC↑ | mAP↑ | AUC↑ |
| | all mechanisms ($h_{\text{cat}}$) | $0.86_{\pm.09}$ | $0.97_{\pm.02}$ | $0.91_{\pm.06}$ | $0.97_{\pm.03}$ | $0.86_{\pm.04}$ | $0.97_{\pm.01}$ | $0.91_{\pm.03}$ | $0.97_{\pm.01}$ |
| Hard | all mechanisms ($h_{\text{diff}}$) | $0.78_{\pm.05}$ | $0.95_{\pm.03}$ | $0.89_{\pm.04}$ | $0.96_{\pm.02}$ | $0.83_{\pm.05}$ | $0.96_{\pm.01}$ | $0.89_{\pm.03}$ | $0.96_{\pm.02}$ |
| | exclude scale | $0.82_{\pm.05}$ | $0.96_{\pm.02}$ | $0.88_{\pm.04}$ | $0.95_{\pm.02}$ | $0.85_{\pm.05}$ | $0.97_{\pm.01}$ | $0.90_{\pm.04}$ | $0.96_{\pm.02}$ |
| | all mechanisms ($h_{\text{cat}}$) | $1.0_{\pm.00}$ | $1.0_{\pm.00}$ | $0.99_{\pm.01}$ | $1.0_{\pm.00}$ | $1.0_{\pm.00}$ | $1.0_{\pm.00}$ | $1.0_{\pm.00}$ | $1.0_{\pm.00}$ |
| Soft | all mechanisms ($h_{\text{diff}}$) | $0.99_{\pm.01}$ | $1.0_{\pm.00}$ | $0.98_{\pm.01}$ | $0.99_{\pm.00}$ | $1.0_{\pm.00}$ | $1.0_{\pm.00}$ | $1.0_{\pm.00}$ | $1.0_{\pm.00}$ |
| | exclude scale | $0.83_{\pm.10}$ | $0.95_{\pm.05}$ | $0.76_{\pm.07}$ | $0.89_{\pm.04}$ | $0.97_{\pm.03}$ | $1.0_{\pm.00}$ | $0.96_{\pm.02}$ | $0.99_{\pm.01}$ |

Table 11: While the majority of classic causal discovery models struggle to scale to larger graphs, we show that CDN still performs reasonably on $N = 100, E = 200$ graphs. AUC reported over 10 i.i.d. graphs.

| Intervention | Linear (1) | Linear (3) | Sigmoid (1) | Sigmoid (3) | Polynomial (1) | Polynomial (3) |
|---|---|---|---|---|---|---|
| Hard | $.64 \pm .04$ | $.65 \pm .08$ | $.63 \pm .02$ | $.61 \pm .06$ | $.50 \pm .01$ | $.49 \pm .03$ |
| Soft | $.92 \pm .01$ | $.90 \pm .03$ | $.72 \pm .02$ | $.73 \pm .02$ | $.53 \pm .03$ | $.52 \pm .05$ |

Table 12: Intervention target prediction results on synthetic datasets, extended results. Uncertainty is standard deviation over 5 i.i.d. datasets. Metrics are averaged over all $3N$ perturbations for a given dataset (1-3 targets). CDN (full) denotes that we trained on all intervention mechanisms jointly, i.e. as in Table 10.

| N | E | Model | Linear (Hard) | | Linear (Soft) | | Poly. (Hard) | | Poly. (Soft) | | Sigmoid. (Hard) | | Sigmoid (Soft) | |
|---|---|---|---|---|---|---|---|---|---|---|---|---|---|---|
| | | | mAP↑ | AUC↑ | mAP↑ | AUC↑ | mAP↑ | AUC↑ | mAP↑ | AUC↑ | mAP↑ | AUC↑ | mAP↑ | AUC↑ |
| 10 | 10 | UT-IGSP | .29±.02 | .57±.03 | .30±.01 | .63±.02 | .33±.04 | .59±.03 | .32±.01 | .64±.02 | .28±.03 | .58±.04 | .26±.02 | .59±.03 |
| | | DCDI-G | .39±.04 | .52±.04 | .40±.04 | .51±.04 | .38±.03 | .49±.02 | .36±.03 | .49±.03 | .37±.03 | .49±.03 | .40±.04 | .52±.06 |
| | | DCDI-DSF | .36±.02 | .51±.04 | .40±.05 | .53±.05 | .37±.02 | .50±.01 | .38±.02 | .49±.03 | .37±.02 | .50±.02 | .37±.02 | .48±.03 |
| | | BACADI-E | .26±.03 | .61±.05 | .42±.13 | .63±.11 | .25±.03 | .61±.05 | .30±.05 | .61±.04 | .28±.04 | .64±.06 | .43±.12 | .69±.09 |
| | | BACADI-M | .23±.01 | .56±.03 | .37±.11 | .62±.11 | .22±.02 | .56±.04 | .30±.05 | .61±.04 | .25±.03 | .61±.05 | .42±.11 | .69±.09 |
| | | DCI | .55±.08 | .79±.04 | .50±.08 | .70±.05 | .47±.09 | .71±.06 | .41±.02 | .67±.01 | .59±.07 | .80±.05 | .51±.05 | .76±.04 |
| | | MB+CI | .53±.05 | .64±.03 | .57±.10 | .66±.06 | .70±.09 | .79±.07 | .85±.07 | .89±.05 | .84±.08 | .88±.06 | .96±.05 | .97±.04 |
| | | CDN | .86±.02 | .93±.01 | .74±.06 | .85±.06 | .83±.09 | .91±.05 | .89±.05 | .94±.03 | .88±.06 | .94±.04 | .85±.07 | .91±.05 |
| | | CDN (no $\mathcal{L}_G$) | .67±.09 | .78±.07 | .82±.05 | .89±.05 | .82±.08 | .91±.05 | .91±.04 | .96±.02 | .89±.05 | .95±.03 | .94±.03 | .97±.02 |
| | | CDN (full) | .87±.01 | .94±.01 | .99±.01 | 1.0±.00 | .82±.08 | .91±.05 | 1.0±.00 | 1.0±.00 | .95±.05 | .98±.02 | 1.0±.00 | 1.0±.00 |
| 10 | 20 | UT-IGSP | .28±.02 | .56±.03 | .26±.01 | .59±.03 | .28±.03 | .58±.04 | .27±.01 | .60±.01 | .25±.00 | .55±.03 | .25±.02 | .56±.05 |
| | | DCDI-G | .42±.02 | .54±.02 | .40±.03 | .51±.04 | .40±.06 | .53±.05 | .39±.02 | .51±.03 | .37±.02 | .50±.01 | .35±.03 | .49±.04 |
| | | DCDI-DSF | .41±.03 | .52±.03 | .39±.05 | .52±.05 | .39±.02 | .51±.02 | .39±.03 | .50±.05 | .36±.03 | .50±.05 | .36±.03 | .49±.03 |
| | | BACADI-E | .34±.04 | .68±.03 | .53±.08 | .71±.04 | .33±.03 | .72±.04 | .64±.07 | .78±.05 | .34±.05 | .71±.04 | .51±.07 | .73±.04 |
| | | BACADI-M | .27±.02 | .63±.03 | .48±.09 | .71±.04 | .27±.04 | .64±.08 | .59±.09 | .77±.06 | .28±.03 | .65±.05 | .45±.08 | .71±.05 |
| | | DCI | .59±.03 | .78±.03 | .57±.04 | .77±.03 | .68±.07 | .82±.04 | .65±.08 | .84±.05 | .70±.03 | .84±.02 | .66±.06 | .83±.05 |
| | | MB+CI | .54±.11 | .66±.09 | .49±.06 | .62±.04 | .83±.04 | .87±.03 | .96±.02 | .96±.02 | .80±.09 | .84±.08 | .96±.02 | .96±.02 |
| | | CDN | .88±.04 | .95±.02 | .56±.07 | .71±.07 | .96±.02 | .98±.01 | .95±.04 | .97±.02 | .94±.04 | .98±.02 | .68±.02 | .79±.02 |
| | | CDN (no $\mathcal{L}_G$) | .79±.06 | .90±.04 | .77±.04 | .86±.02 | .94±.01 | .97±.01 | .95±.05 | .96±.04 | .87±.02 | .93±.02 | .94±.02 | .96±.01 |
| | | CDN (full) | .89±.03 | .94±.02 | .98±.02 | .99±.01 | .97±.01 | .99±.00 | 1.0±.00 | 1.0±.00 | .96±.03 | .98±.01 | .98±.02 | .99±.01 |
| 20 | 20 | UT-IGSP | .15±.01 | .54±.01 | .18±.01 | .65±.02 | .20±.02 | .60±.02 | .21±.02 | .67±.02 | .15±.01 | .57±.02 | .17±.01 | .65±.01 |
| | | DCDI-G | .24±.02 | .49±.02 | .22±.02 | .49±.02 | .22±.03 | .50±.03 | .22±.02 | .49±.03 | .22±.02 | .50±.04 | .22±.02 | .50±.03 |
| | | DCDI-DSF | .24±.02 | .50±.02 | .23±.02 | .50±.03 | .22±.01 | .50±.02 | .20±.01 | .48±.02 | .22±.02 | .50±.04 | .21±.02 | .47±.03 |
| | | BACADI-E | .14±.01 | .61±.03 | .25±.10 | .60±.06 | .22±.06 | .76±.06 | .30±.06 | .74±.05 | .15±.02 | .65±.03 | .17±.05 | .57±.05 |
| | | BACADI-M | .12±.00 | .58±.02 | .17±.03 | .60±.05 | .14±.02 | .66±.05 | .22±.05 | .71±.06 | .12±.01 | .59±.04 | .14±.02 | .56±.05 |
| | | DCI | .43±.03 | .77±.01 | .45±.08 | .75±.04 | .48±.05 | .76±.03 | .50±.02 | .79±.02 | .46±.05 | .75±.02 | .38±.12 | .73±.06 |
| | | MB+CI | .48±.09 | .66±.05 | .57±.08 | .70±.05 | .72±.10 | .84±.06 | .92±.04 | .94±.02 | .68±.06 | .80±.04 | .89±.03 | .92±.02 |
| | | CDN | .74±.02 | .90±.01 | .85±.06 | .96±.02 | .78±.09 | .92±.04 | .92±.02 | .98±.00 | .75±.06 | .91±.03 | .78±.06 | .92±.04 |
| | | CDN (no $\mathcal{L}_G$) | .50±.08 | .73±.07 | .87±.03 | .96±.01 | .76±.09 | .90±.05 | .92±.03 | .98±.01 | .71±.04 | .89±.02 | .93±.02 | .99±.00 |
| | | CDN (full) | .75±.03 | .91±.02 | .99±.01 | 1.0±.00 | .79±.08 | .93±.04 | 1.0±.00 | 1.0±.00 | .76±.05 | .92±.02 | 1.0±.00 | 1.0±.00 |
| 20 | 40 | UT-IGSP | .14±.01 | .57±.01 | .14±.01 | .61±.02 | .20±.01 | .62±.02 | .19±.01 | .65±.01 | .13±.01 | .56±.02 | .14±.00 | .60±.01 |
| | | DCDI-G | .22±.03 | .49±.03 | .21±.02 | .48±.03 | .22±.02 | .49±.02 | .22±.02 | .50±.02 | .21±.01 | .48±.02 | .23±.01 | .51±.02 |
| | | DCDI-DSF | .21±.02 | .46±.03 | .21±.02 | .47±.02 | .23±.02 | .50±.03 | .21±.01 | .49±.03 | .21±.02 | .49±.02 | .21±.03 | .49±.02 |
| | | BACADI-E | .23±.05 | .70±.04 | .33±.10 | .67±.04 | .29±.08 | .83±.04 | .45±.18 | .78±.06 | .20±.05 | .73±.05 | .51±.13 | .79±.04 |
| | | BACADI-M | .15±.01 | .64±.02 | .19±.04 | .63±.02 | .17±.02 | .70±.05 | .31±.10 | .77±.07 | .14±.01 | .65±.03 | .32±.06 | .77±.04 |
| | | DCI | .60±.06 | .85±.03 | .50±.02 | .77±.03 | .56±.08 | .81±.05 | .59±.06 | .84±.04 | .64±.06 | .87±.02 | .66±.05 | .88±.03 |
| | | MB+CI | .53±.09 | .70±.06 | .47±.06 | .64±.03 | .78±.04 | .88±.02 | .95±.03 | .96±.02 | .83±.03 | .89±.02 | .96±.04 | .97±.03 |
| | | CDN | .86±.04 | .96±.02 | .79±.09 | .93±.05 | .88±.03 | .97±.01 | .96±.02 | .99±.00 | .88±.03 | .97±.01 | .71±.05 | .87±.04 |
| | | CDN (no $\mathcal{L}_G$) | .69±.09 | .85±.07 | .80±.06 | .94±.03 | .84±.03 | .94±.01 | .95±.02 | .99±.00 | .84±.05 | .93±.03 | .91±.02 | .98±.00 |
| | | CDN (full) | .89±.06 | .97±.02 | .99±.01 | 1.0±.00 | .89±.03 | .97±.01 | 1.0±.00 | 1.0±.00 | .93±.03 | .98±.01 | 1.0±.00 | 1.0±.00 |

