# OpenReview forum: "Identifying biological perturbation targets through causal differential networks"
_ICML.cc/2025/Conference — ICML 2025 poster_

### Official Review · Reviewer_JJyS · 2025-03-15

**Overall Recommendation:** 2

**Summary:**

The authors considered the problem of identifying the target of interventions from merely observational/interventional data. In some biological applications, there might be thousands of variables in the system and few samples are available per intervention. The authors utilized an amortized approach (which has been explored for the task of causal discovery in the literature of causality) to detect the target of interventions. In particular, a network called causal structure learner is trained to return the causal graph from the observed data. Then, a differential network is considered where the graph obtained from observational data and the one from the interventional data are passed to an attention-based classifier to detect which nodes are intervened on. Both the causal structure learner network and the differential network are trained jointly with amortized data collected from different causal models and interventions. The experimental results showed that the proposed method performs better than previous work.

### Update after rebuttal
I have read the reviews and the rebuttal. In my view, the proposed approach remains quite heuristic and lacks theoretical grounding. That said, the experimental results are promising. Overall, I still recommend a weak reject, though I acknowledge it could be accepted.

**Claims And Evidence:**

The work is experimental and in the experiment section, the authors made a comprehensive comparison with previous work.

**Essential References Not Discussed:**

As far as I checked the related work, the authors cited the most relevant previous work.

**Experimental Designs Or Analyses:**

I did not check the codes, but I read the experiment section. Based on the text, it seems that the experimental results are sound and the authors also provided some explanations for the plots.

**Methods And Evaluation Criteria:**

The authors considered several metrics for comparison such as rank, Spearman rank correlation, and Pearson correlation. Moreover, they considered several datasets in their evaluations.

**Other Comments Or Suggestions:**

- It would be nice to study whether some theoretical guarantees can be added to the work in some specific settings. Moreover, it is good to check how this method performs in more challenging cases such as under latent confounding.

**Other Strengths And Weaknesses:**

Strengths:
- The authors used attention-based networks and proposed an architecture to learn the location of interventions with an amortized scheme.
- They compared with several previous works in different synthetic and real datasets and showed better performance in almost all the datasets.

Weakness:
- The main weakness is that there is no theoretical guarantee in the work. That being said, I should emphasize that this comment is also applied to some of the previous work. Nevertheless, it is interesting to see whether, in simple settings, such as linear models or bivariate cases, the proposed method indeed recovers the correct intervention targets.

**Questions For Authors:**

- Regarding the causal structure learner, what are the alternatives instead of SEA? Why did the authors consider this architecture?
- The text in lines 193-198 is ambiguous. How is $E'$ formed from the output of FCI? What are $k$ and $T$ in this context?
- Since FCI uses conditional independence (CI) tests, in the regime considered in this paper with thousands of variables, the output of FCI might have too many errors. It is not clear how the proposed method performs well with such a noisy input. Moreover, the correlation matrix does not encode any CI assertions (which are essential in learning the equivalence class).
- I did not understand the sentence "we randomly sample a permutation of the maximum graph size to each graph" about the edge embedding. It is not clear how this ensures that the embedding is equivariant to the labeling of nodes. Moreover, why is this approach considered in this paper?

**Relation To Broader Scientific Literature:**

One of the main applications of causal inference/discovery is in biology such as learning causal structures in gene regulatory networks or predicting the results of gene perturbations. This work is aligned with this line of research and aims to find the target of interventions (such as protein targets of a drug).

**Theoretical Claims:**

The work is experimental and there is no theoretical claim in the paper.

---

> ### Author Rebuttal · Authors · 2025-03-31
>
> Thank you for your comments! Please let us know if this helps answer your questions, and if you have any further concerns.
>
> > Theoretical guarantees
>
> We show in Appendix C that the differential network is *well-specified* as a *model class*. Due to the complexity of the architecture, it is difficult to *guarantee* what the model will learn, i.e. in the identifiability sense. At the same time, it is *not* a given that an arbitrary Transformer can approximate any arbitrary function given limited precision, width, depth [1,2]. We are inspired by the neural network expressivity literature to show that (page 15):
>
> - Hard interventions: given the sets of edges $E, E'$, where $E'$ represents the mutilated graph, there exists a setting of its parameters that maps $(E, E')$ to targets $I$ by detecting edges that disappeared.
> - Soft interventions: under "scale" interventions and non-multiplicative noise, there exists a setting of its parameters that maps the correlation and covariance matrices (which can be derived from model inputs) to targets $I$. In the linear case, Eq. 14-15 naturally hold and are relevant to the relationships between variables (i.e. if $x$ is an intervention target, for every $y\ne x$, correlation($x,y$) does not change while cov($x,y$) does).
>
> [1] Yun et al. Are Transformers universal approximators of sequence-to-sequence functions? ICLR 2020
>
> [2] Sanford et al. Representational Strengths and Limitations of Transformers. NeurIPS 2023.
>
> > Alternative causal structure learners
>
> CSIvA [1] and AVICI [2] are supervised causal discovery algorithms that could work in principle.
>
> - CSIvA does not have a public codebase or supplemental files.
>
> - AVICI is available and runs well on smaller datasets, but there are some practical challenges. Since the Transformer attends over raw data, its complexity scales quadratically as the number of samples. Also, AVICI was only trained on datasets of up to 50 variables and 200 samples. On graphs of 100 variables (Table 7 in [2]), the performance drops to that of classical algorithms (GES, GIES).
>
> We chose [3] since it reported reasonable performance on larger datasets (CausalBench, 622 variables). The use of summary statistics like correlation also mirrors common practices in single cell sequencing, since the raw data are often very noisy (150-152, right).
>
> [1] Ke et al. Learning to Induce Causal Structure. ICLR 2023.
>
> [2] Lorch et al. Amortized inference for causal structure learning. NeurIPS 2022.
>
> [3] Wu et al. Sample, estimate, aggregate: A recipe for causal discovery foundation models. TMLR 2025.
>
> > lines 193-198
>
> Thank you for pointing this out, and please let us know if this makes more sense.
>
> FCI is run $T$ times on different subsets of $k$ nodes. The "space of possible edge types" (196) refers to the edge types output by FCI, which are mapped to categorical labels (e.g. no edge to 0, -> to 1, – to 2). These elements are "embedded" similar to word embeddings in language models (201-202).
>
> So $E'$ contains $k\times k \times T$ elements ($T$ adjacency matrices, each of size $k\times k$).
>
> > FCI noisy estimates and correlation
>
> Our causal graph learner, finetuned from SEA [3], uses correlation and FCI estimates and as input features. Correlation does not encode CI information, but it may help the model quickly filter out edges that are *not* likely to exist, especially important for larger datasets.
>
> FCI is run on subsets of $k=5$ throughout (above), rather than the full dataset [194]. These smaller subsets are sampled based on heuristics like high pairwise correlation, to prioritize CI testing for regions likely to be "interesting."
>
> There are classic algorithms for integrating (sub)graph estimates over different variable sets [4]. SEA trains a neural network to perform this task over datasets of varying assumptions, precisely because FCI estimates may be noisy in practice [3]. In this sense, the role of the neural network is to "denoise" the graph prediction and to learn to resolve contradictions.
>
> [4] Huang et al. Causal discovery from multiple data sets with non-identical variable sets. AISTATS 2020.
>
> > Permuting the edge embedding
>
> This practice follows SEA [3] and loosely resembles the idea in [5], where permutations are sampled at random during training so that "in expectation" the Transformer does not rely on any particular ordering.
>
> It's necessary to add embeddings to inform of node/edge identities across layers, since Transformers are permutation invariant. In principle, any ordering will do. However, since synthetic datasets are generated in topological order, "node 1" always corresponds to a root node and so on. As [6] shows, it's trivial for neural networks to memorize this spurious feature to recover topological order.
>
> [5] Yang et al. XLNet: Generalized Autoregressive Pretraining for Language Understanding. NeurIPS 2019.
>
> [6] Reisach et al. Beware of the Simulated DAG! Causal Discovery Benchmarks May Be Easy To Game. NeurIPS 2021.

---

### Official Review · Reviewer_smYK · 2025-03-16

**Overall Recommendation:** 3

**Summary:**

The authors focus on the problem of identifying direct targets of intervention in single-cell perturbation data. The proposed method, Causal Differential Network (CDN), first extracts causal networks behind unperturbed and perturbed data. These networks are then compared using an axial attention-based classifier to determine which nodes (genes) are direct targets of the intervention. The method was evaluated on multiple single-cell sequencing screens involving gene and drug perturbations, as well as synthetic datasets.

**Claims And Evidence:**

Yes, the performance results reported by the authors in the paper (e.g., Tables 1 and 3) adequately demonstrate the effectiveness of the proposed CDN method. The results of the ablation experiments are also sound.

**Essential References Not Discussed:**

None that I'm aware of.

**Experimental Designs Or Analyses:**

Refer to Claims And Evidence and Methods And Evaluation Criteria.

**Methods And Evaluation Criteria:**

The authors conducted experiments on five commonly used perturbation datasets and several synthetic datasets. The dataset selection and evaluation metrics appear reasonable.

**Other Comments Or Suggestions:**

Please refer to Other Strengths And Weaknesses.

**Other Strengths And Weaknesses:**

The novelty of this paper is debatable. It appears to be a direct application of existing CDN algorithms to the target identification task, with minimal contribution to the machine learning algorithm aspect. The biological aspect also lacks more interpretable analyses (such as phenotype prediction or exploration of genetic interactions).

**Questions For Authors:**

Please refer to Other Strengths And Weaknesses.

**Relation To Broader Scientific Literature:**

This paper combines the amortized causal discovery model[1] with the axial attention-based classifier[2], improving performance on the target identification task in perturbation datasets like [3], surpassing previously common models for this task like [4].

[1] Wu, Menghua, et al. "Sample, estimate, aggregate: A recipe for causal discovery foundation models." arXiv preprint arXiv:2402.01929 (2024).
[2] Ho, Jonathan, et al. "Axial attention in multidimensional transformers." arXiv preprint arXiv:1912.12180 (2019).
[3] Dixit, Atray, et al. "Perturb-Seq: dissecting molecular circuits with scalable single-cell RNA profiling of pooled genetic screens." cell 167.7 (2016): 1853-1866.
[4] Roohani, Yusuf, Kexin Huang, and Jure Leskovec. "Predicting transcriptional outcomes of novel multigene perturbations with GEARS." Nature Biotechnology 42.6 (2024): 927-935.

**Theoretical Claims:**

This paper does not contain relevant theoretical proofs.

---

> ### Author Rebuttal · Authors · 2025-03-31
>
> Thank you for your comments! Please let us know if this response helps clarify your questions, and if you have any further concerns.
>
> > "direct application of existing CDN algorithms to the target identification task"
>
> CDN is the name of the method we propose; are you referring to causal discovery algorithms?
>
> Our key contribution is the idea of supervised causal discovery (predicting graphs from simulated data) as a pretraining objective, to obtain dataset representations useful for downstream tasks like inferring targets.
>
> A direct solution to the task is to predict targets without inferring graphs. However, we empirically demonstrate that graph prediction leads to substantial improvement as an auxiliary loss (Table 3, no $L_G$) and starting from pretrained weights leads to more stable optimization dynamics (Figure 10). Furthermore, baselines that use data-mined graph priors perform substantially worse (Table 1, Figure 3).
>
> We realize our idea with [1] as the causal structure learner, and a 2D Transformer-based architecture as the differential network, *neither of which* are our primary contributions. These modules were selected for their scalability and could (in principle) be replaced with alternate supervised learning algorithms [2,3] or models that operate pairs of graph [4].
>
> [1] Wu et al. Sample, estimate, aggregate: A recipe for causal discovery foundation models. 2024.
>
> [2] Ke et al. Learning to Induce Causal Structure. ICLR 2023.
>
> [3] Lorch et al. Amortized inference for causal structure learning. NeurIPS 2022.
>
> [4] Li et al. Graph Matching Networks for Learning the Similarity of Graph Structured Objects. ICML 2019.
>
> > interpretable biological analyses
>
> This work focuses on the application of predicting perturbation targets, for efficient experimental design. The goal is *not to replace* experiment via simulation, but to *lower* the cost/scale of experiments, by reducing the search space. We do not believe that this focus diminishes from the interpretability of the results.
>
> - Since the causal graph learner is supervised on synthetic graphs, the predicted graphs are always available for inspection. This is not a primary focus of this work, due to the lack of cell line-specific "ground truth" graphs for evaluation. However, the predicted edges represent gene-gene interactions, and the differences between the observational and interventional graphs correspond to differences in gene regulation.
>
> - Works that do predict phenotypes often share our intended application [5], and as we write in the introduction, directly predicting the target is more efficient at inference time, especially for combinations.
>
> - Directly predicting targets also has benefits in terms of label / evaluation quality. Perturbation target identity is one of the most reliable readouts from these single-cell data, as it is *non*-quantitative [6]. In contrast, due to technical and biological covariates, two datasets generated by the same lab, of the same cell line, can be largely inconsistent in terms of individual gene expression. For example, [7] writes of K562 genome-wide and essential: "The median correlation between  log-fold-change point estimates [...] was only 0.16, suggesting very low replicability."
>
> [5] Roohani et al. GEARS: Predicting transcriptional outcomes of novel multi-gene perturbations. Nat Biotech 2024.
>
> [6] Luecken and Theis. Current best practices in single‐cell RNA‐seq analysis: a tutorial. Mol Syst Biol (2019) 15: e8746.
>
> [7] Nadig et al. Transcriptome-wide characterization of genetic perturbations. 2024.

---

> > ### Comment · Reviewer_smYK · 2025-04-01
> >
> > I think the authors’ response clearly articulated their motivation and the inspiration behind their proposed method, so I have raised my score.

---

> > > ### Author Response · Authors · 2025-04-01
> > >
> > > Thank you for your response and for updating your score! Your comments are very helpful for improving our paper.

---

### Official Review · Reviewer_ryKm · 2025-03-17

**Overall Recommendation:** 4

**Summary:**

Identifying targets that induce cell state transitions has numerous implications in advancing targeted drug discovery and biological research. Authors propose causal differential networks (CDN), an attention-based framework independent of biological priors to identify targets of an intervention given a pair of observational and interventional datasets. The CDN framework is comprised of two modules – (1) a causal structure learner that predicts causal graphs that could have produced each dataset and (2) a differential network classifier that predicts targets of an intervention from the causal graphs – that are jointly trained via a supervised amortized framework. The paper benchmarks CDN on 7 real transcriptomic datasets containing genetic (Perturb-seq) and chemical (Sci-Plex) perturbations and shows that CDN outperforms the state-of-the-art perturbation modeling methods for both seen and unseen cell lines. Experiments on synthetic datasets show that CDN outperforms existing causal discovery methods on predicting intervention targets. Overall, the paper presents a data-driven framework to identify perturbation targets, a task that can help elucidate large-scale perturbation experiments, causal targets, and targeted drug discovery.

**Claims And Evidence:**

Most are reasonably supported in the paper. It was unclear how the model handled the noisiness in the data and also if the synthetic data adequately represented the underlying properties of the real-world datasets.

**Essential References Not Discussed:**

Most relevant work has been discussed

**Experimental Designs Or Analyses:**

The experimental section is quite detailed

**Methods And Evaluation Criteria:**

- The proposed methods make sense – the paper uses a causal learner and differential network causal discovery framework to identify perturbation targets of interventions using a data-driven approach and does not rely on biological knowledge graph priors that may potentially be confounding.

- Benchmark datasets make sense for the task –  the paper includes 5 Perturb-seq datasets for seen and unseen cell line settings, 2 chemical perturbation datasets, and synthetic datasets.

- However, in the evaluation criteria, the paper uses the normalized rank of the true target, and it is unclear how the rank is computed. There is some additional information regarding ranking candidates in Appendix B.2 but only for GEARS and GENEPT.

**Other Comments Or Suggestions:**

N/A

**Other Strengths And Weaknesses:**

Strengths:

- The originality of the work lies in the novel combination of a causal structure learner and differential networks.
The significance of the work is substantial, as it is a purely data-driven method for identifying perturbation targets and outperforms existing methods.

- The evaluation framework includes a robust collection of benchmarking datasets and baselines.
Weaknesses:

- The paper states that the causal learner is largely adapted from the architecture of SEA (Wu et al.) but does not describe the architecture of SEA. What differences and similarities exist between the casual learner architecture of SEA and CDN?

- The clarity of the paper could be improved. The paper mentions that datasets are represented “in terms of their summary statistics” but does not state exactly which statistics. In Methods 3.1 Causal structure learner (Input featurization), authors explain that global statistics “like pairwise correlation” are computed, while Methods 3.3 Implementation details (Model parameters) mention that the global statistic is a pairwise correlation. Is pairwise correlation the only type of global statistic computed?

- The rank evaluation criteria could also be further explained.

**Questions For Authors:**

Covered above

**Relation To Broader Scientific Literature:**

In the broader scientific literature, many machine learning based methods exist to identify perturbation targets. CDN demonstrates great potential for target discovery, as it outperforms state-of-the-art perturbation modeling approaches. The significance of the proposed method lies in the data-driven causal framework to identify perturbation and causal targets, which has extensive applications to biological research and targeted therapeutic development.

**Theoretical Claims:**

I did not check the theoretical claims carefully.

---

> ### Author Rebuttal · Authors · 2025-03-31
>
> Thank you for your comments! Please let us know if this response helps clarify your questions, and if you have any further concerns.
>
> > Architecture similarities and differences / novelty
>
> Our key contribution is the idea of supervised causal discovery (predicting graphs from simulated data) as a pretraining objective, to obtain dataset representations useful for downstream tasks like inferring targets. We realize our idea with [1] as the causal structure learner, and a 2D Transformer-based architecture as the differential network, neither of which are our primary contributions. These modules were selected for their scalability and could (in principle) be replaced with alternate supervised learning algorithms [2,3] or models that operate pairs of graph [4].
>
> For a specific comparison to [1] –
>
> Similarities: We take the aggregator in SEA as a pretrained module, which is finetuned.
>
> Differences:
> - We finetune SEA using correlation as a summary statistic. The original paper uses the precision matrix instead (not suitable for single-cell data), but writes that it's possible to finetune weights for different statistics.
> - The differential network is separate and learned from scratch.
> - We generate our own datasets for training throughout.
>
> [1] Wu et al. Sample, estimate, aggregate: A recipe for causal discovery foundation models. TMLR 2025.
>
> [2] Ke et al. Learning to Induce Causal Structure. ICLR 2023.
>
> [3] Lorch et al. Amortized inference for causal structure learning. NeurIPS 2022.
>
> [4] Li et al. Graph Matching Networks for Learning the Similarity of Graph Structured Objects. ICML 2019.
>
> > Data noise and relevance of synthetic data
>
> We agree that real data are noisy, hence the focus on summary statistics (correlation, mean, variance) as the primary inputs to the model.
>
> Simulating mRNA expression is a task on its own, and improvements in that regard are directly applicable here. In this work, the synthetic soft interventions were loosely inspired by perturbations, e.g. in terms of fold-changes (line 796).
>
> > Summary statistics
>
> We'll make the inputs more clear. Thank you for pointing this out.
>
> Yes, the causal structure learner uses pairwise correlations as the summary statistic. In addition to the paired graph features, the differential network takes marginal means and variances (218-219 right, inspired by differential expression analysis).
>
> > Rank criteria
>
> We'll be more explicit regarding the definition.
>
> For perturbation effect predictors (GEARS, GenePT), we order genes based on the similarity of their predicted effect to the ground truth target effect (Eq 3, B.2). Methods that directly predict targets order genes based on their predicted probability of being a target. The set of candidate genes is the set of top 1000 differentially-expressed genes.
>
> The rank (309-311) is the index of the correct answer in any ordering, divided by the number of candidates (since different perturbations have different candidate pools, B.2), where 1 is best (top of light) and 0 is worst (bottom of list).
>
> This is equivalent to the expected recall at p (Table 3), over all p.
>
> Rank is inspired by the application of experimental design. Since it is common to select top predictions for further testing, "rank" quantifies how many perturbations must be tested on average, before finding the ground truth.

---

> > ### Comment · Reviewer_ryKm · 2025-04-03
> >
> > Thank you for clarifying my questions. Just a quick follow-up question - was there any sensitivity analysis done for the choice of top-n differential expressed genes? How was the 1000 chose? Was it based on existing literature?

---

> > > ### Author Response · Authors · 2025-04-03
> > >
> > > Thank you for your response!
> > >
> > > We chose the top 1000 based on adjusted p-value (details in D.1), which is quite common as a selection criteria + a way to rank genes [1, 2]. Specifically, this is the p-value of testing – are the pre and post perturbation distributions the same – adjusted for multiple testing. 1000 was somewhat of an arbitrary threshold, loosely inspired by the number of HVGs people typically choose + the limit for finetuning the model on a single GPU.
> > >
> > > - For example, [3] writes: "In a genome-scale Perturb-seq screen, we find that a typical gene perturbation affects an estimated 45 genes, whereas a typical essential gene perturbation affects over 500 genes."
> > >
> > > - In addition, [4] writes regarding highly-variable gene selection (global selection across all perturbations, based on variance to mean ratio): "Depending on the task and the complexity of the dataset, typically between 1,000 and 5,000 HVGs are selected for downstream analysis [...]. Preliminary results from Klein et al (2015) suggest that downstream analysis is robust to the exact choice of the number of HVGs. While varying the number of HVGs between 200 and 2,400, the authors reported similar low‐dimensional representations in the PCA space."
> > >
> > > - Finally, a classic approach for identifying drug targets [5] also samples 1000 candidate targets as decoys.
> > >
> > > [1] Love et al. Moderated estimation of fold change and dispersion for RNA-seq data with DESeq2. Genome Biol 15, 550 (2014)
> > >
> > > [2] Heumos et al. Best practices for single-cell analysis across modalities. Nat Rev Genet 24, 550–572 (2023). – "P values obtained with DGE tests over conditions must be corrected for multiple testing to obtain q values."
> > >
> > > [3] Nadig et al. Transcriptome-wide characterization of genetic perturbations. 2024.
> > >
> > > [4] Luecken and Theis. Current best practices in single‐cell RNA‐seq analysis: a tutorial. Mol Syst Biol (2019) 15: e8746.
> > >
> > > [5] Isik et al. Drug target prioritization by perturbed gene expression and network information. Scientific Reports 2015.
> > >
> > > The statistical test doesn't inherently induce uncertainty, so we ran differential expression analysis five times on 80% of the samples. The selection of differentially expressed genes is quite consistent "at the top" and varies slightly lower down the list.
> > >
> > > Specifically, the top 20 DE genes in each split based on adjusted p-value are present in other splits 95% of the time. The predictive performance of predicting the true target changes very little (Table 7, standard deviations from 0 to 0.02 across splits).
> > >
> > > The top 50 DE genes are present in other splits 92% of the time, and the top 1000 are present 79% of the time.

---

### Official Review · Reviewer_K5QP · 2025-03-18

**Overall Recommendation:** 3

**Summary:**

This paper introduces Causal Differential Networks (CDN), a method to identify which variables in a biological system are directly perturbed between observational and interventional conditions. The authors frame the problem in terms of causal discovery: they first learn approximate causal graphs from each condition, then use an attention-based neural network to detect which nodes have shifted causal mechanisms, thus identifying the perturbation targets. The pipeline is trained end-to-end in an amortized fashion using both simulated and real single-cell transcriptomics data.
Experimental results suggest that this approach achieves superior accuracy in identifying causal perturbation targets compared to the baselines, both on large-scale real datasets of genetic and chemical perturbations and on carefully controlled synthetic datasets.

**Claims And Evidence:**

The claims made in the paper are generally supported by empirical evidence, but I have the following concerns:

*  Motivation and Practical Significance: The paper should more clearly justify the motivation and practical significance of identifying perturbation targets across different types of perturbation data.

   * In chemical perturbation experiments, where precise molecular targets are often unknown, identifying perturbation targets is intuitive and valuable. However, in genetic perturbation experiments, where genes are directly targeted, the value lies in distinguishing direct vs. downstream effects and aiding in combinatorial perturbations. The significance of the proposed method is not fully discussed, leaving readers unclear about its impact.

   * The paper's main contribution is identifying perturbation targets from observational and interventional data, but the authors do not demonstrate its practical use. It is unclear how the proposed method can be utilized. For instance, can it help learn a better causal model? Can it improve combinatorial perturbation prediction? Or is it meant to guide experimental design? These potential applications are not clearly articulated.

* Interpretability and Biological Significance: From an interpretability perspective, the biological significance of the inferred causal relationships is not explored. This leaves questions about whether the model captures meaningful biological mechanisms, which is crucial for validating its utility in real-world biological research.

**Essential References Not Discussed:**

All the essential related works are discussed, but it would be better to include works on single-cell foundation models as the perturbation prediction is one of the most important downstream tasks.

**Experimental Designs Or Analyses:**

The experimental design and analysis are generally well-strctured and sound.

**Methods And Evaluation Criteria:**

The proposed model architecture and the evaluation criteria make sense to me.

**Other Comments Or Suggestions:**

* Please see Other Strengths And Weaknesses.

* (Minor typos) The citation of “Rao et al. (2021)” at lines 579–584 on the left column of page 11 needs to be revised.

**Other Strengths And Weaknesses:**

Weaknesses:

* please see my comments on the previous parts.

* Zero-Shot Generalization Concerns: The authors claim the proposed model can generalize to unseen perturbations, but it's unclear whether it can perform good zero-shot inference. Recent work by [1] has shown that the identifiability of amortized causal discovery methods depends heavily on the training data. While the authors mention generating 8640 datasets with hard and soft interventions, they provide insufficient details on how these datasets are simulated. The paper would benefit from a detailed discussion on generalization ability under distribution shift, especially if the prepared dataset is heterogeneous.

* Theoretical Foundation: The paper lacks theoretical analysis regarding the identifiability conditions under which the model is guaranteed to correctly identify perturbation targets. Such analysis would strengthen the theoretical underpinnings of the approach.


[1] Montagna, F., Cairney-Leeming, M., Sridhar, D., & Locatello, F. (2024). Demystifying amortized causal discovery with transformers. arXiv preprint arXiv:2405.16924.

**Questions For Authors:**

Please see Other Strengths And Weaknesses.

**Relation To Broader Scientific Literature:**

This paper bridges two key research domains: single-cell perturbation modeling and multi-environment causal discovery. While single-cell biology typically focuses on predicting intervention outcomes or mapping perturbations to known pathways without directly inferring causal structure, multi-environment causal discovery methods leverage varied conditions to enhance identifiability but often fail to scale to high-dimensional single-cell data.
The paper's key contribution is an amortized causal inference framework that learns local causal graphs from both observational and interventional data without relying on external knowledge networks. By analyzing differences between these graphs, the method identifies perturbation targets in noisy, high-dimensional single-cell gene expression datasets. This integration of approaches addresses limitations in both fields, offering a solution for causal inference in complex biological systems.

**Theoretical Claims:**

The paper does not introduce new proofs or theorems. However, it would benefit from a theoretical analysis on the identifiability guarantees of the proposed model. Specifically, it is important to understand under what conditions the model is guaranteed to accurately identify perturbation targets.

---

> ### Author Rebuttal · Authors · 2025-03-31
>
> Thank you for your comments! Please let us know if this response helps clarify your questions, and if you have any further concerns.
>
> > Intended application
>
> We will make this more explicit in the paper. This work focuses on the application of predicting perturbation targets, for efficient experimental design. The goal is not to replace experiment via simulation, but to lower the cost/scale of experiments, by reducing the search space.
>
> We envision for actual use cases to resemble the chemical perturbation case more, e.g.
> - Inputs: before/after drug (unknown mechanism)
> - Model generates hypotheses for molecular mechanisms of drugs
> - Top predictions should be verified experimentally, i.e. knock down proposed genes and compare to "after drug" phenotype.
>
> This type of method has an advantage to phenotype prediction methods when considering combinations of targets, e.g. messy drugs. Different off-targets may contribute orthogonal effects, which may make the predicted phenotype of any single target less similar to the overall profile. In addition, there are limited real training data for combinatorial perturbations, motivating methods that can leverage synthetic data.
>
> > Why Perturb-seq?
>
> Some experiments use Perturb-seq datasets because in terms of model evaluation, the ground truth targets are relatively "clean." CRISPR is very specific compared to drugs, which may impact many proteins, known and unknown. While this is not the most "realistic" application, we chose these datasets for their size and ease of evaluation.
>
> > Single cell foundation models
>
> We will include discussion of these works. Thank you for the suggestion!
>
> > Generalization under distribution shift
>
> Our "unseen cell line" case resembles "zero-shot" generalization, since experiments in different cell lines are necessarily conducted separately (282). While these datasets were produced by the same lab, they are actually quite incomparable at the level of individual genes, even for the same cell line. For example, [1] writes of K562 genome-wide and essential: "The median correlation between log-fold-change point estimates [...] was only 0.16"
>
> [1] Nadig et al. Transcriptome-wide characterization of genetic perturbations. 2024.
>
> In controlled settings, our synthetic experiments hold out the "scale" soft intervention from training (368 right, Table 9). We include the exact formula for these interventions / causal mechanisms in D.2 (p16-17).
>
> > Interpretability of intermediate graphs
>
> Since the causal graph learner is supervised on synthetic graphs, the predicted graphs are always available for inspection. This is not a primary focus of this work, due to the lack of cell line-specific "ground truth" graphs for evaluation. However, the predicted edges could represent gene regulatory relationships, and the differences between the observational and interventional graphs correspond to differences in gene regulation.
>
> > Theoretical analysis
>
> We show in Appendix C that the differential network is *well-specified* as a *model class*. Due to the complexity of the architecture, it is difficult to *guarantee* what the model will learn, i.e. in the classical identifiability sense. At the same time, it is *not* a given that an arbitrary Transformer can approximate any arbitrary function given limited precision, width, depth, [2,3]. Instead, we are inspired by the neural network expressivity literature to show that (page 15):
>
> - Hard interventions: given the sets of edges $E, E'$, where $E'$ represents the mutilated graph, there exists a setting of its parameters that maps $(E, E')$ to targets $I$ by detecting edges that disappeared.
> - Soft interventions: under "scale" interventions and non-multiplicative noise, there exists a setting of its parameters that maps the correlation and covariance matrices (which can be derived from model inputs) to targets $I$. In the linear case, Eq. 14-15 naturally hold and are relevant to the relationships between variables (i.e. if $x$ is an intervention target, for every $y\ne x$, correlation($x,y$) does not change while cov($x,y$) does.
>
> [2] Yun et al. Are Transformers universal approximators of sequence-to-sequence functions? ICLR 2020
>
> [3] Sanford et al. Representational Strengths and Limitations of Transformers. NeurIPS 2023.
>
> > Typos
>
> Thank you for pointing this out! We will fix it.

---

### Official Review · Reviewer_gPde · 2025-03-18

**Overall Recommendation:** 4

**Summary:**

The paper proposes a new method called causal differential networks (CDN) for identifying variables responsible for biological perturbations (e.g. drug targets) from pairs of observational and interventional datasets. The main idea involves first inferring noisy causal graphs from data, and then learning a mapping from these causal graphs to the set of variables subjected to intervention. The authors demonstrate that CDN significantly outperforms existing methods across seven single-cell transcriptomics datasets and synthetic benchmarks.

**Claims And Evidence:**

I believe the claims made in the submissions are supported by convincing evidence (up to some reservations on the experimental setting).

**Essential References Not Discussed:**

To my knowledge, all the essential references are discussed.

**Experimental Designs Or Analyses:**

I have some doubts regarding the experimental setting:

- The authors restrict the target search to the top 1000 differentially expressed genes. This choice can bias the evaluation because if the true perturbation target is already among those top candidates, even a simple baseline like differential gene expression (DGE) can appear very competitive. It’s not entirely clear whether this candidate restriction reflects a realistic use case or if it artificially favors methods that exploit strong differential signals.
- The synthetic experiments are conducted on random graphs with only 10–20 nodes, which is a much simpler setting than the high-dimensional real transcriptomic data. Given the scalability of the proposed method, I would expect the authors to evaluate their approach on larger graphs.
- The training and test splits are created by clustering perturbations based on log-fold change profiles. It would be useful if the authors conducted a sensitivity analysis—comparing results across various clustering methods (e.g., different k values in k-means or random initializations) to demonstrate that their conclusions hold regardless of how the data is partitioned.

**Methods And Evaluation Criteria:**

Yes the proposed methods and benchmark datasets are very relevant to the application at hand (target prediction).

**Other Comments Or Suggestions:**

N/A

**Other Strengths And Weaknesses:**

N/A

**Questions For Authors:**

- Is restricting the target search to the top 1000 differentially expressed genes a standard practice in the literature?
- How sensitive are the results to the choice of clustering algorithm or the number of clusters (e.g., different k values in k-means)?

**Relation To Broader Scientific Literature:**

The key contributions are well discussed in the context of the broader scientific literature.

**Theoretical Claims:**

There are no theoretical claims in the paper.

---

> ### Author Rebuttal · Authors · 2025-04-01
>
> Thank you for your comments! Please let us know if these answers are helpful, and if you have any further concerns.
>
> > Top 1000 DEGs
>
> It's quite common to restrict analysis to subsets of genes, though the threshold and criteria may vary. Here, we believe that selecting the top 1000 genes (based on differential expression p-value) is reasonably permissive, since these genes are selected per perturbation.
>
> For example, [1] writes: "In a genome-scale Perturb-seq screen, we find that a typical gene perturbation affects an estimated 45 genes, whereas a typical essential gene perturbation affects over 500 genes."
>
> In addition, [2] writes regarding highly-variable gene selection (global selection across all perturbations, based on variance to mean ratio): "Depending on the task and the complexity of the dataset, typically between 1,000 and 5,000 HVGs are selected for downstream analysis [...]. Preliminary results from Klein et al (2015) suggest that downstream analysis is robust to the exact choice of the number of HVGs. While varying the number of HVGs between 200 and 2,400, the authors reported similar low‐dimensional representations in the PCA space."
>
> Finally, a classic approach for identifying drug targets [3] argues that targets are not highly differentially expressed, but network-based approaches are useful for predicting them from gene expression. There, they also sample 1000 candidate targets as decoys.
>
> [1] Nadig et al. Transcriptome-wide characterization of genetic perturbations. 2024.
>
> [2] Luecken and Theis. Current best practices in single‐cell RNA‐seq analysis: a tutorial. Mol Syst Biol (2019) 15: e8746.
>
> [3] Isik et al. Drug target prioritization by perturbed gene expression and network information. Scientific Reports 2015.
>
> > Larger synthetic experiments
>
> We are happy to include experiments on larger graphs ($N=100, E=200$, AUC over 10 new datasets). There is some drop in performance, especially Polynomial. This could be attributed to error propagation from SEA, which performs less well (at predicting graphs) on larger graphs, and could potentially be ameliorated by finetuning on larger synthetic data (similar to how Perturb-seq models were finetuned to be larger).
>
> Linear + Soft remains quite good, perhaps because the theory (Appendix C) suggests that the summary statistics are sufficient for predicting targets, without the graph prediction.
>
> | Intervention | Linear (1) | Linear (3) |  Sigmoid (1) | Sigmoid (3) | Polynomial (1) | Polynomial (3) |
> |--|--|--|--|--|--|--|
> | Hard | .64±.04 | .65±.08 | .63±.02 | .61±.06 | .50±.01 | .49±.03 |
> | Soft |  .92±.01 |.90±.03 | .72±.02 |  .73±.02 | .53±.03 | .52±.05
>
> > Different test splits
>
> Typically, the largest cluster (assigned to train, e.g. nearly 1/3 of K562) contains weak perturbations with minimal phenotype, so different clusterings don't introduce as much heterogeneity.
>
> Instead, we can partition all perturbations uniformly into 5 sets for the unseen cell line setting. As seen here for K562 genome-wide, the variation across sets is minimal. We will include similar results for all cell lines.
>
> | Split | Rank | $\rho$ | $r$ | Recall-1 | Recall-5 | Recall-20
> |--|--|--|--|--|--|--|
> | Original Test (n=678) | 0.974 | 0.915 | 0.921 | 0.699 | 0.900 | 0.957 |
> | New splits (n=353 x5)| 0.973 | 0.917 | 0.923 | 0.696 | 0.908 | 0.961 |
> | Std of above | 0.009 | 0.017 | 0.016 | 0.031 | 0.032 | 0.021

---

### Official Review · Reviewer_mWJN · 2025-03-19

**Overall Recommendation:** 2

**Summary:**

This paper proposes two modules for predicting true perturbation targets of cell perturbation datasets. Firstly, a causal structure learner is pre-trained on synthetic data in an attempt to recover causal graph representation from both observational and interventional dataset. Then, a differential network is trained on both synthetic data and downstream real-world cellular data to predict the intervention targets.

**Claims And Evidence:**

The model architectures look pretty clear and straightforward to me, which is good. However, for the same reason, I'm not quite sure where the novelty lies. For causal structure learner, the authors stated that the proposed method is based on [1], then what is the difference of the causal structure learner in this work compared to [1]? And how does this difference improves on [1]? I don't think the novelty & contribution of this work on top of prior works is stated very clearly. [1] seems like an unpublished work and I don't think it is particularly well-known and well-accepted in the causal discovery community to the point that you can omit stating your contribution in comparison to it.


[1] Wu,M., Bao,Y., Barzilay,R., and Jaakkola,T. Sample, estimate, aggregate: A recipe for causal discovery foundation models. arXiv2402.01929,2024.

**Essential References Not Discussed:**

Key prior work [1] that outperformed CPA with convincing margins should be cited and discussed on L64-67 right side.

[1] Wu, Yulun, et al. "Predicting cellular responses with variational causal inference and refined relational information." ICLR (2023).

**Experimental Designs Or Analyses:**

The soundness and validity of experimental designs are fine to me.

**Methods And Evaluation Criteria:**

The causal structure learner is quite heuristic and theoretically loose. It is generally well-known that identifying causal structure from observational dataset alone is not possible [1] and some additional data / prior knowledge / assumptions need to be present [2][3]. I do not believe pre-training the causal structure learner on unrelated synthetic dataset with true synthetic causal graph is going to help model's ability to identify causal representation on observational cellular datasets in the 2nd module, unless the synthetic dataset contains some true dynamics about cell perturbations. Therefore, I think $h_{obs}$ is just a graph representation that contains correlation information about the features in the observational dataset, not causal information. Besides, why is double edge allowed as an edge type (L196 left side)? Is causal structure not represented as directed acyclic graph?


[1] Locatello, Francesco, et al. "Challenging common assumptions in the unsupervised learning of disentangled representations." international conference on machine learning. PMLR, 2019.

[2] Brehmer, Johann, et al. "Weakly supervised causal representation learning." Advances in Neural Information Processing Systems 35 (2022): 38319-38331.

[3] Shen, Xinwei, et al. "Weakly supervised disentangled generative causal representation learning." Journal of Machine Learning Research 23.241 (2022): 1-55.

**Other Comments Or Suggestions:**

1. I think the authors could be more clear in terms of details on equations and notations. For example, it is not quite clear to me how intervention target is inferred by the equation and descriptions on L75-80 right side, since the notations are not clearly defined and explained. I can only assume that given a cell, it is the intervention whose empirical response distribution on the dataset is the most correlated with this cell's expression profile? Besides, $k$ and $T$ on L195-198 left side is also quite confusing to me as there is no explicit explanations of what they are. For differential network, there is only a figure and verbal explanations regarding the mechanism but no exact equations on the attention and aggregation mechanisms. I managed to find the attention mechanism in Appendix C but it is not pointed to by the main text.

2. The conclusion section is missing.

**Other Strengths And Weaknesses:**

Strengths:

Good evaluation results are shown for the proposed method.

**Questions For Authors:**

Questions are embedded in the previous sections.

**Relation To Broader Scientific Literature:**

This work is related to prior work that uses deep learning to predict intervention targets on cell perturbation dataset, such as [1]. The authors pointed out that it is also related to prior work that performs cellular response prediction such as [2], [3], through which intervention target can be implicitly inferred. More broadly, it is related to prior work in causal discovery and causal representation learning as the ones I cited in the "Methods And Evaluation Criteria" section.


[1] Gonzalez, Guadalupe, et al. "Combinatorial prediction of therapeutic perturbations using causally-inspired neural networks." bioRxiv (2024).

[2] Lotfollahi, Mohammad, et al. "Predicting cellular responses to complex perturbations in high‐throughput screens." Molecular systems biology 19.6 (2023): e11517.

[3] Hetzel, Leon, et al. "Predicting cellular responses to novel drug perturbations at a single-cell resolution." NeurIPS (2022).

**Theoretical Claims:**

No theoretical claims.

---

> ### Author Rebuttal · Authors · 2025-03-31
>
> Thank you for your comments! Please let us know if this response helps clarify your questions, and if you have any further concerns.
>
> > Technical novelty
>
> Our key contribution is the idea of supervised causal discovery (predicting graphs from simulated data) as a pretraining objective, to obtain dataset -> graph representations useful for downstream tasks like inferring targets. A direct solution to the task is to predict targets without inferring graphs. However, we empirically demonstrate that graph prediction leads to substantial improvement as an auxiliary loss (Table 3, no $L_G$) and starting from pretrained weights leads to more stable optimization dynamics (Figure 10). Furthermore, baselines that use data-mined graph priors perform substantially worse (Table 1, Figure 3).
>
> We realize our idea with [1] as the causal structure learner, and a 2D Transformer-based architecture as the differential network, *neither of which* are our primary contributions. These modules were selected for their scalability and could (in principle) be replaced with alternate supervised learning algorithms [2,3] or models that operate pairs of graphs [4].
>
> [1] Wu et al. Sample, estimate, aggregate: A recipe for causal discovery foundation models. 2024.
>
> [2] Ke et al. Learning to Induce Causal Structure. ICLR 2023.
>
> [3] Lorch et al. Amortized inference for causal structure learning. NeurIPS 2022.
>
> [4] Li et al. Graph Matching Networks for Learning the Similarity of Graph Structured Objects. ICML 2019.
>
> > Identifiability
>
> Our setting is slightly different from the typical causal representation learning setting, where high-dimensional observations (e.g. images) are associated with (unobserved) low-dimensional factors, on which interventions are applied.
>
> We follow the classic causality framework (Sec 2.1), where we measure all variables (causal sufficiency), and interventions are applied directly to these variables. Many classical causal discovery algorithms apply to the observational setting, in which certain graph topologies are still identifiable, under various assumptions [5].
>
> Supervised causal discovery algorithms do not have explicit assumptions, but they do follow implicit ones captured by the training data [6].
>
> [5] Sprites, Glymour, and Scheines. Causation, prediction, and search. 2001.
>
> [6] Montagna et al. Demystifying amortized causal discovery with transformers. 2024.
>
> > Value of synthetic data.
>
> Empirically, the baselines that predict targets from real data without external information / synthetic data do not perform as well as our full method (Table 1). The synthetic soft interventions were loosely inspired by perturbations, e.g. in terms of fold-changes (796). In general, synthetic data pretraining also improves training stability (Figure 10).
>
> > Bidirected edges
>
> The model's input and output spaces are different. The causal graph learner outputs only three edge types: "A -> B," "B -> A," "no edge." FCI graph estimates are *inputs* to the model, along with correlation. The input edge type embedding includes bidirected edges because FCI may estimate them.
>
> > Prior work
>
> Thank you for the recommendation! We will include this in the discussion of related work.
>
> > L75-80 right
>
> These lines describe how a perturbation effect "simulator" could be used to identify targets. The actual implementation for GEARS and GenePT is described in B.2. We will update this section to be more descriptive and add a reference.
>
> - "space of perturbations $\mathcal{I}$" refers to the set of sets of perturbation targets $I$ that are possible, e.g. the powerset of all variables, if we allow interventions of arbitrary # of targets
> - "an oracle $\phi : D_\text{obs}, I \to P$ refers to the simulator, which takes as input, an observational dataset (starting cells) and a set of intervention targets (genes)
> - "the $I^*$ associated with $\tilde{P*}$ refers to the set of ground truth targets, associated with the target distribution (Eqs 1 and 2)
> - "dissimilarity d" could be cosine distance, or mean squared error, or KL divergence between distributions
> - $\arg\min$ over $\mathcal{I}$ implies an optimization over all possible sets of targets, e.g. naively we run the model $\phi$ once per $I$ and output the $I$ whose prediction is closest to the target cells.
>
> > What are k and T?
>
> We will update the paper to clarify. FCI is run $T$ times on different subsets of $k$ nodes to produce subgraph estimates (input to the causal graph learner)
>
> > Differential network
>
> The differential network architecture is implemented using standard multi-head attention, applied along each axis. We will expand the descriptions in the main text and include references to the appropriate details.
>
> > Conclusion is missing
>
> We will add a conclusion to highlight the key contributions, limitations, and future directions.

---

> > ### Comment · Reviewer_mWJN · 2025-04-03
> >
> > Thank the authors for the response, particularly the explanations. I understand that there is a tight character limitation for rebuttal and no allowance for revision paper upload, so it is hard to elaborate the revisions in detail. Could the authors use the reply to this comment to post your updated content for *"What are k and T?"*, *"Prior work"*, and *"Conclusion is missing"*? If I find the updated content clear and comprehensive, and if I have your word that these content will be in the revised version of the paper, I'll raise my score. Otherwise, I wouldn't be so confident, since some large chunk of content such as conclusion and limitations are entirely missing.

---

> > > ### Author Response · Authors · 2025-04-03
> > >
> > > Thank you for your response! We will definitely include this updated content in our revised paper (we have already modified our copy with these and other edits), and we appreciate the time spent by all reviewers in providing helpful feedback.
> > >
> > > Please let us know if these sections convey what you felt was missing.
> > >
> > > > k and T
> > >
> > > 3.1 Input Featurization
> > >
> > > First, we compute global statistics $\rho\in R^{N\times N}$ (pairwise correlation) over all variables. **This statistic is an inexpensive way for the model to filter out edges that are not likely to occur.** Second, we obtain estimates of local causal structure **to orient edges**, by running classical causal discovery algorithms over small subsets of variables. **Specifically, we run the FCI algorithm T times over subsets of k variables (Spirtes et al., 1995), which are sampled proportional to their pairwise correlation.** We denote the set of estimates as $E' \in E^{k\times k\times T}$ , where E is the space of possible edge types **inferred by FCI** (e.g. ↔, −, →). **The size and number of local estimates**, k and T , are hyperparameters that trade off accuracy vs. runtime **(Section 3.3)**
> > >
> > > Section 3.3 Implementation Details
> > >
> > > Marginal estimates E′ are computed using the classic FCI algorithm (Spirtes et al., 1995) with **k = 5 and T = 100**
> > >
> > > > Prior work
> > >
> > > L64-67 right
> > >
> > > Alternatively, models like GEARS (Roohani et al., 2023), **graphVCI (Wu et al., 2023)**, AttentionPert (Bai et al., 2024), and GIM (Schneider et al., 2024) infer the effects of unseen perturbations. These latter methods generalize by modeling the relationships between seen and unseen perturbations. For example, GEARS leverages literature-derived relationships in the Gene Ontology knowledge graph (Ashburner et al., 2000), and **graphVCI learns to refine the gene regulatory relationships implied by ATAC-seq data**
> > >
> > > > Conclusion
> > >
> > > This work introduces causal differential networks (CDN) for predicting perturbation targets in the context of single-cell transcriptomics. Our key technical contribution is the idea of using supervised causal discovery as a pretraining objective, to obtain dataset representations useful for downstream tasks like inferring targets. Specifically, CDN trains a *causal graph learner* that maps a pair of observational and interventional datasets to their data-generating mechanisms, and a *differential network* classifier to identify variables whose conditional independencies have changed. We demonstrated that our approach outperforms state-of-the-art methods in perturbation modeling and causal discovery, across seven transcriptomics datasets and a variety of synthetic settings. Overall, these results highlight the potential of our method to design more efficient experiments, by reducing the search space of perturbations.
> > >
> > > There are several limitations of this work and directions for future research. First, our implementation does not address cyclic, feedback relationships or time-resolved, dynamic systems – both of which are common in biology. Modeling these cases may require data simulation schemes that can sample from the steady-state of a cyclic system, or architectures that directly predict the parameters of a system of stochastic differential equations (Lorch et al., 2024). In addition, current supervised causal discovery algorithms, which we use for the causal graph learner, tend to assume causal sufficiency, i.e. that there are no unseen confounders. This assumption is unrealistic in biology, as it is not physically possible to measure all relevant variables. Thus, there is an opportunity to extend this work for the latent confounding setting. Finally, since it is difficult to simulate biology, there will be an inevitable domain shift between synthetic and real data. However, there does exist plentiful unlabeled single-cell data (Consortium* et al., 2022), which could be used with self-supervised objectives to bridge the gap (Sun et al., 2020). In conclusion, we hope that this work will enable efficient experimental design and interpretation, as well as future machine learning efforts towards these tasks.
> > >
> > > [1] Lorch et al. Causal modeling with stationary diffusions. AISTATS 2024.
> > >
> > > [2] The Tabula Sapiens: A multiple-organ, single-cell transcriptomic atlas of humans. Science 376, eabl4896 (2022).
> > >
> > > [3] Sun et al. Test-time training with self-supervision for generalization under distribution shifts. ICML 2020.

---

### Official Review · Reviewer_1GJy · 2025-03-20

**Overall Recommendation:** 4

**Summary:**

The paper introduces causal differential networks (CDN), a supervised method designed to identify which variables (eg. genes) were directly targeted by biological perturbations (genetic knockouts or chemical treatments) using gene expression data from single-cell transcriptomics experiments. CDN integrates a causal graph inference model (based on the SEA model architecture) and a differential perturbation classifier into a joint end-to-end framework. The causal graph inference module estimates causal graphs from observational and perturbed conditions separately; a subsequent attention-based differential module identifies intervention targets by comparing inferred causal graphs and related statistics (eg. mean, covariance, correlation).

CDN employs a Transformer-based architecture with specialized axial attention layers to handle graph data in a permutation-invariant manner, allowing scalability and robustness. Training involves pre-training on synthetic causal graphs and fine-tuning on real single-cell transcriptomics data. CDN was tested on five genetic perturbation datasets (perturb-seq) and two chemical perturbation datasets (sci-plex). The model is shown to outperform sota causal discovery and deep-learning baselines on real genetic perturbations and showed reasonable success in the more challenging chemical perturbation setting.

The main findings demonstrate that CDN achieves better identification accuracy than prior methods, generalizes well to unseen cell lines with minimal drop in performance, and handles both hard (knockouts) and soft (scaling) interventions effectively. The paper provides a good experimental evaluation on real and synthetic data to validate CDN's performance.

**Claims And Evidence:**

The paper's key claims, e.g. superior accuracy in identifying intervention targets, robustness to unseen biological domains, and improved performance over traditional causal discovery algorithms, are well supported by multiple experiments on real single-cell datasets and carefully designed synthetic benchmarks. CDN consistently ranks the true targets higher and shows stronger correlation between predicted and actual intervention effects compared to multiple sota baselines.

I think the claim of generalization is especially convincing, backed by experiments explicitly evaluating CDN's performance on unseen cell lines, showing minimal performance drop. Similarly, CDN's ability to detect multiple simultaneous targets and handle different intervention types (hard and soft) is well supported by synthetic experiments. I believe the authors appropriately interpret results without exaggeration or unsupported claims.

**Essential References Not Discussed:**

I think the paper could benefit by further contextualizing contributions in bioinformatics works on perturbation signature matching, earlier and more recent differential network approaches. Including these references would provide a bit historical context, emphasizing how CDN builds upon past methods and distinguishes from recent ones.

1. **Lamb et al. (2006). "The Connectivity Map: using gene-expression signatures to connect small molecules, genes, and disease."**
This work introduced the idea of connecting small molecule drugs to gene targets (and diseases) via gene expression signatures​. It laid the groundwork for using transcriptomic comparative analysis to infer drug targets (e.g. if a drug's expression profile is similar to that of a gene knockout, that gene might be the drug's target)​.

2. **Noh & Gunawan (2016). "Inferring gene targets of drugs and chemical compounds from gene expression profiles".**
I think this work is directly relevant, as it proposed DeltaNet, a method to infer drug targets by solving a regression problem linking expression changes to putative targets (without needing to first infer a full network)​. It outperformed network-based methods like MNI and SSEM, showing the value of a direct data-driven approach​. DeltaNet is, in a sense, a precursor to CDN but using a linear model and least-angle regression instead of a learned neural network​.

3. **Wang et al. (2018). "Direct estimation of differences in causal graphs."** I believe the DCI algorithm used in the cited work (Belyaeva
et al., 2021) stems from this paper from Wang et al.

4. **Chen et al. (2023). "iSCAN: identifying causal mechanism shifts among nonlinear additive noise models."** Similar setting to DCI but extends to nonlinear additive models.

5. **Malik et al. (2024). "Identifying causal changes between linear structural equation models."** Similar to DCI, the authors study the linear setting for difference networks but provide more analyses on the graphical conditions leading to identifiability of the intervention targets as well as the difference edges. Also related to the cited work (Varici et al., 2022).

**Experimental Designs Or Analyses:**

The experimental design is appropriate, and well executed in my opinion. Data preprocessing, perturbation candidate selection, and train-test splits were carefully described and justified. Filtering trivial perturbations and ensuring splits by perturbation-effect clustering help prevent overfitting and leakage.

Metrics chosen are appropriate for the task. The authors carefully assessed uncertainty through repeated runs or bootstrapping. CDN's advantage over baselines was large enough that minor statistical variations did not affect conclusions. The design explicitly tested CDN’s ability to generalize across cell lines and to handle different perturbation types, with thoughtful analyses like ablations and runtime measurements.

**Methods And Evaluation Criteria:**

The proposed CDN methodology, including the causal graph inference and perturbation classifier modules, is well justified for the problem domain. Using a combined graph representation and statistical features to capture intervention effects is sensible, given biological perturbation complexity. The authors provide clear details about model architecture, training, validation, and evaluation metrics.

The evaluation strategy is detailed, comparing CDN against a good range of baselines, including both domain-specific (eg. GEARS, PDGRAPHER) and general causal inference methods. Real data seem to be carefully preprocessed, with sensible candidate filtering and careful splitting to prevent data leakage. Metrics chosen (normalized rank, correlation, recall@p, and synthetic mAP/AUC) are standard and appropriate for capturing model accuracy.

**Other Comments Or Suggestions:**

None

**Other Strengths And Weaknesses:**

### Strengths:

* Original integration of causal structure learning with neural network prediction.
* Rigorous experimental validation, detailed ablation and robustness analysis.
* Good clarity, strong statistical robustness, transparent details aiding reproducibility.
* Real-world biological application with clear potential impact.

### Weaknesses:

* Dependence on synthetic data realism and labeled perturbation targets (maybe potential domain adaptation challenge).
* Implicit assumption of DAG structure; handling cycles not explicitly addressed.
* Limited interpretability of intermediate causal graphs.

**Questions For Authors:**

I would appreciate if the authors can comment on these questions:

1. How does your method handle situations where the true causal structure is not acyclic? Biological pathways often contain feedback loops or reciprocal interactions. In such cases, I image CDN's performance might largely drop and a dynamic or time-series approach would be needed. How challenging is extending CDN's to dynamic causal graphs?

2. Your training strategy heavily relies on simulated data to teach the model causal patterns. How sensitive is CDN to the choice of simulation parameters and assumptions? For instance, if the real data had a fundamentally different type of causal dynamics (e.g. non-linearities not in your simulation), would CDN struggle? Did you observe any mismatch issues when fine-tuning on real data? If one cannot perfectly simulate a domain, how would you recommend mitigating simulation-to-reality gaps?

3. CDN is trained to output a set of targets. In your synthetic experiments it handled multiple simultaneous interventions. In real-world cases like drug perturbations, a drug can have multiple targets (on-target and off-target proteins). How well would CDN handle identifying multiple true targets? Is it capable of assigning high scores to more than one gene if multiple are responsible for the observed changes? If not directly, what modifications would be needed (e.g. a different loss or thresholding criterion) to allow multi-target identification?

**Relation To Broader Scientific Literature:**

I think the authors clearly position its contributions, distinguishing CDN's supervised, amortized causal discovery approach from earlier unsupervised causal discovery methods (UT-IGSP, DCDI, BACADI) and network-based or knowledge-driven perturbation target methods (GEARS, PDGRAPHER). CDN is appropriately recognized as building upon concepts from differential causal network analysis (e.g. DCI), extending them into high-dimensional, non-linear single-cell data.

**Theoretical Claims:**

The paper provides theoretical justifications rather than formal proofs. I (lightly) checked some key derivations ( Appendix C) showing how interventions affect graph structure and global statistics (correlation and covariance) and found them sound.

---

> ### Author Rebuttal · Authors · 2025-04-01
>
> Thank you for your questions and recommendations! We hope this response addresses your questions, but please let us know if you have any additional concerns.
>
> > Additional historical context
>
> Thank you for these references and suggestions. We will incorporate them into the paper alongside additional discussion of the historical context.
>
> > Extension to dynamical systems
>
> An empirical solution is to simulate cyclic data, and to use a classic discovery algorithm for cyclic graphs.
>
> We can also model the dynamical system. In perturbation data, cells are sequenced after perturbations penetrate completely, e.g. 6-8 days [1], so we might assume that the data are sampled from the stationary density of a system of stochastic differential equations [2]. Here, the causal structure learner could predict the SDE parameters instead of the causal graph. Similar to the acyclic case, interventions could be "read off" the differences in the SDE parameters, e.g. $f$ and $\sigma$ in [2]. The key design choice/challenge would be to select an input featurization that allows for the targets to be identified.
>
> For actively differentiating cells (instead of pre/post differentiation), steady state assumptions might not hold. While there exist measurements like RNA velocity (the ratio of spliced to unspliced mRNA) that can partially order cells within a population and inform of causal structure [3, 4], modeling these continuous dynamics is out of the scope of this work.
>
> [1]. Replogle et al. Mapping information-rich genotype-phenotype landscapes with genome-scale Perturb-seq. Cell. 2022.
>
> [2] Lorch et al. Causal Modeling with Stationary Diffusions. AISTATS 2024.
>
> [3] Singh et al. Causal gene regulatory analysis with RNA velocity reveals an interplay between slow and fast transcription factors. Cell Systems 2024.
>
> [4] Atanackovic et al. DynGFN: Towards Bayesian Inference of Gene Regulatory Networks with GFlowNets. NeurIPS 2023.
>
> > Sensitivity to simulation
>
> - CDN is not sensitive to the type of non-linearity among graphs of the same size, as performance remains about the same for novel non-linearities (Polynomial, Table 3 and Sigmoid, Table 10).
> - CDN is more sensitive to the type of intervention (Table 9), so this is a key parameter to understand / simulate.
> - Mismatch issues: During development, we found that finetuning on Perturb-seq was not helpful for the Sci-Plex data, presumably because CRISPR perturbations are much cleaner and stronger than drugs. We did design the synthetic datasets with less specific / soft interventions in mind [below], which may render them a better fit.
>
> **Sim-to-real gap:** If a model has been trained and one would like to simulate additional finetuning data, one could be informed by the data-driven graph connectivity (e.g. based on pairwise correlation) and classic hypothesis tests (e.g. for data distribution, differences between pre/post intervention).
>
> However, these heuristics cannot fully capture complex systems like biology. Instead, perhaps one could use a self-supervised objective to finetune the model on observational data, a la test-time training approaches [5]. Alternatively, one could incorporate biological knowledge graphs with the understanding that they are *noisy* priors. E.g. as an additional input feature or as a finetuning objective for the causal graph learner.
>
> [5] Sun et al. Test-Time Training with Self-Supervision for Generalization under Distribution Shifts. ICML 2020.
>
> > Interpretability of intermediate graphs
>
> Since the causal graph learner is supervised on synthetic graphs, the predicted graphs are always available for inspection. This is not a primary focus of this work, due to the lack of cell line-specific "ground truth" graphs for evaluation. However, the predicted edges could represent gene regulatory relationships, and the differences between the observational and interventional graphs correspond to differences in gene regulation.
>
> > Multi-target identification
>
> Currently, CDN is trained to make an independent prediction per gene (sigmoid, not softmax, 242-244). The synthetic experiments contain up to 3 targets per graph, and the model does output multiple high scores (Table 3).
>
> A limitation of this current approach is that the per-gene decision does not consider other genes in the proposed subset. To (efficiently) consider the entire subset, the prediction could be made autoregressively, where the order of predictions could be based on effect size, or be sampled at random during training.

---

### Decision · Program_Chairs · 2025-05-01

**Decision:**

Accept (poster)

**Comment:**

This paper introduces Causal Differential Networks (CDN), a supervised method that integrates causal graph inference with perturbation target prediction to identify variables directly affected by biological interventions using single-cell gene expression data.

Pros:

+ Uses permutation-invariant axial attention and a well-structured integration of causal inference and perturbation classification.

+ Shows minimal performance drop on unseen biological domains and handles both simple and complex intervention types.

Cons:

+ Causal graph inference relies on pre-training from synthetic data, raising concerns about its ability to learn true causal relationships in real-world scenarios.

+ Evaluation may be biased by restricting target candidates and using small synthetic graphs, which may not reflect real data complexity.